# Transition to siblinghood causes a substantial and long-lasting increase in urinary cortisol levels in wild bonobos

**Verena Behringer[1,2]\*[†], Andreas Berghänel[3†], Tobias Deschner[4], Sean M Lee[5], Barbara Fruth[6,7], Gottfried Hohmann[2,6]**

[1]Endocrinology Laboratory, German Primate Center, Leibniz Institute for Primate Research, Göttingen, Germany; [2]Max Planck Institute for Evolutionary Anthropology, Leipzig, Germany; [3]Domestication Lab, Konrad Lorenz Institute of Ethology, Department of Interdisciplinary Life Sciences, University of Veterinary Medicine Vienna, Vienna, Austria; [4]Comparative BioCognition, Institute of Cognitive Science, University of Osnabrück, Osnabrück, Germany; [5]Center for the Advanced Study of Human Paleobiology, Department of Anthropology, George Washington University, Washington, United States; [6]Max Planck Institute of Animal Behavior, Konstanz, Germany; [7]Centre for Research and Conservation, Royal Zoological Society of Antwerp, Antwerp, Belgium

**\*For correspondence:**
VBehringer@dpz.eu

[†]These authors contributed equally to this work

**Abstract** In animals with slow ontogeny and long-term maternal investment, immatures are likely to experience the birth of a younger sibling before reaching maturity. In these species, the birth of a sibling marks a major event in an offspring's early life as the older siblings experience a decrease in maternal support. The transition to siblinghood (TTS) is often considered to be stressful for the older offspring, but physiological evidence is lacking. To explore the TTS in wild bonobos, we investigated physiological changes in urinary cortisol (stress response), neopterin (cell-mediated immunity), and total triiodothyronine (T3, metabolic rate), as well as changes in behaviors that reflect the mother–offspring relationship. Following a sibling's birth, urinary cortisol levels of the older offspring increased fivefold, independent of their age, and remained elevated for 7 months. The cortisol level increase was associated with declining neopterin levels; however, T3 levels and behavioral measures did not change. Our results indicate that the TTS is accompanied by elevated cortisol levels and that this change does not coincide with nutritional weaning and attainment of physical independence. Our results suggest that bonobos and humans experience TTS in similar ways and that this developmental event may have emerged in the last common ancestor.

## Editor's evaluation

This article examines the behavioral and physiological responses of wild bonobos to the birth of a younger sibling. The findings contribute to our understanding of the effects of a major life history transition in a primate species that is closely related to humans. An important strength of this article is the novel use of a longitudinal dataset that incorporates both behavioral and physiological measures.

## Introduction

In mammals, weaning refers to the transition from nutritional dependency to a stage when immatures are independent of maternal food provisioning. The term weaning is often used for the attainment

of nutritional independence, but also comprises the process of social independence and behavioral maturation, which can occur at different ages. Weaning age varies within and across species, and is an important developmental stage in the life history of mother and offspring (**Smith, 2013**; **Weary et al., 2008**). While the dependency on post-weaning maternal support can be inferred from behavioral observations, the putative fitness effects are rarely explored. A reduction or complete loss of maternal support has substantial fitness costs throughout an individual's life span (**Zipple et al., 2021**). However, while maternal loss is a dramatic event, there are normative events such as sibling birth that affect the life of the older offspring. In vertebrate species with a slow development, many immatures grow up with siblings, and sibling relationships can have profound influences on fitness (**Berger et al., 2021**; **Nitsch et al., 2013**). The younger sibling may benefit from an older sibling in terms of survival, reproductive maturation, and socialization (**Berger et al., 2021**; **Nitsch et al., 2013**; **Stanton et al., 2017**). However, the older offspring must share maternal care, which may influence its social behavior as well as its physiological constitution.

Primates differ from most other social mammals in having remarkably slow life histories (**Charnov and Berrigan, 1993**; **Jones, 2011**). Immatures grow slowly, social maturation extends well into adulthood, and to a certain degree, beneficial mother–offspring relationships can last a lifetime (**Jones, 2011**; **Pereira and Fairbanks, 1993**; **Surbeck et al., 2019**). Therefore, female primates may give birth to another infant before the older offspring reaches physical or social maturity, or even before being weaned. For the older offspring, this transition to siblinghood (TTS) marks the onset of considerable changes, including the sudden emergence of a competitor for maternal resources (sibling rivalry; **Dettwyler, 2017**; **Myers and Bjorklund, 2018**) and a decline in maternal support (**Kramer, 2011**). Accordingly, in humans, TTS is considered to be a stressful life event for the older sibling even under favorable conditions, a perspective that seems to be supported by TTS-related behaviors of the older offspring such as aggression, clinginess, and depressive syndromes. However, sibling birth also presents opportunities for the older offspring, such as social and emotional growth through interacting with the newborn. Individuals vary in how they adjust to the birth of a younger sibling; some children have difficulties while others cope well (reviewed in **Volling, 2012**; **Volling et al., 2017**). In any case, the birth of a sibling is linked to a time of change the older child must cope with. Evidence from nonhuman primates is scarce, but the available information resembles reports from humans (**Devinney et al., 2003**; **Schino and Troisi, 2001**). However, whether behavioral changes during TTS are actually associated directly with sibling birth, or are rather simply a result of age-related withdrawal of maternal support, remains to be resolved (**Volling, 2012**; **Volling et al., 2017**).

TTS could overlap with and/or accelerate weaning and attainment of physical independence, which, on its own, is known to be stressful in primates and other mammals (e.g., **Hau and Schapiro, 2007**; **Mandalaywala et al., 2014**). As a result, it is difficult to differentiate between the effects of sibling birth and weaning (**Volling, 2012**; **Weary et al., 2008**). Nutritional weaning refers to the termination of an offspring's consumption of maternal milk, though they may still continue nipple contact (without milk transfer) – this is assumed to be a social comfort behavior (**Bădescu et al., 2017**; **Berghänel et al., 2016**; **Matsumoto, 2017**). It is common that females give birth to another infant before the older offspring reaches full independence, resulting in an overlap of dependency in siblings of different ages (**Achenbach and Snowdon, 1998**). In nonhuman primates, sibling birth affects the quality and quantity of interactions between the older offspring and the mother (**Schino and Troisi, 2001**; **van Noordwijk and van Schaik, 2005**), and may affect the fitness of the older offspring throughout its life (**Alberts, 2019**; **Bădescu et al., 2022**; **Thompson et al., 2016**; **Tung et al., 2016**; **Zipple et al., 2019**).

Apes offer a particularly suitable model to explore developmental changes in an evolutionary context (**Sayers, 2015**): maternal support is intense and persists for a long time (**Stanton et al., 2020**; **van Noordwijk et al., 2018**), and extended periods of parental care of two dependent offspring of different ages are common (**Achenbach and Snowdon, 1998**). Juvenile apes associate with their mother for several years after nutritional weaning. While data on mother–offspring relationships are abundant, little is known about the interactions between immatures and infants born to the same female (**Watts and Pusey, 1993**). Wild orangutans have the longest known mammalian inter-birth interval (7–9 years), and sibling rivalry is likely modest or less intense since the close association of the mother with the older offspring ends before the next infant is born (**van Noordwijk et al., 2018**; **van Noordwijk and van Schaik, 2005**). In gorillas, inter-birth intervals range from 4 to 6 years (**Stoinski**

*et al., 2013*). In male mountain gorillas, sibling bonds may last into adulthood (*Robbins, 1995*), and following maternal loss, siblings may provide social support (*Morrison et al., 2021*), indicating that siblings are strong partners in this species. In wild chimpanzees, inter-birth intervals range from 2 to 11 years (*Thompson, 2013*). There is one anecdotal report of an older offspring responding to sibling birth with increasing attempts to establish physical contact with the mother and the emergence of signs of depression (*Clark, 1977*). Based on this, it can be assumed that depending on the species, immature apes experiencing the birth of a sibling are exposed to different social environments: Because a greater difference in sibling ages may correspond to less conflict in terms of their maternal support needs, it is likely that species with shorter inter-birth intervals (gorillas and chimpanzees) experience stronger effects of TTS than those with longer intervals (orangutans).

Immature bonobos depend heavily on their mothers and maintain close spatial and physical contact during the first 2 years of life (*De Lathouwers, 2004*; *Kuroda, 1989*; *Lee et al., 2020*). After the age of 5 years, spatial distance to the mother increases (*Kuroda, 1989*; *Toda et al., 2021*), but in the case of sons, associations between mothers and offspring persist even when sons reach adulthood (*Hohmann et al., 1999*; *Surbeck et al., 2019*). Nutritional weaning occurs between 4 and 5 years of age (*Kuroda, 1989*; *Oelze et al., 2020*), and behavioral observations and urinary cortisol measures indicate that nutritional weaning is less stressful in bonobos than in chimpanzees (*De Lathouwers and Van Elsacker, 2006*; *Tkaczynski et al., 2020*). Notably, monitoring changes in urinary cortisol levels during weaning revealed the first evidence that the older offspring may respond physiologically to the birth of a sibling (*Tkaczynski et al., 2020*).

Here, we investigate TTS-related changes in physiological responses in wild habituated juvenile bonobos (*Pan paniscus*) at LuiKotale in the Democratic Republic of Congo. We used multiple physiological and behavioral measures to investigate the responses of older siblings to the birth of their younger sibling. We sought to disentangle the effects of changes in mother–offspring relationships and energetics that are associated with nutritional and social weaning, from the specific effects of a younger sibling's birth. We leveraged the large variation in inter-birth intervals in bonobos (*Knott, 2001*; *Tokuyama et al., 2021*) to differentiate between the effects of TTS versus nutritional and social weaning. In our study population, inter-birth intervals ranged from 2.3 to 8.6 years (mean ± SD = 5.4 ± 1.5 years) (*Tkaczynski et al., 2020*), and thus the developmental status of older siblings at the time when their mothers gave birth to another infant ranged from highly dependent in terms of travel support and foraging skills (i.e., time carried and nursed) to mostly independent. This discordance between inter-birth interval lengths and the developmental timelines of the older offspring enabled us to explore the specific effects of TTS on older siblings' (a) physiological stress response (cortisol), (b) immunity (neopterin), (c) energetic change (total T3), (d) relationships with their mothers, and (e) changes in foraging and travel competence, while controlling for (f) offspring sex and age.

Changes in cortisol are widely accepted as a physiological marker to quantify stress responses in humans and other mammals because after exposure to a stressor – an event that challenges homeostasis – cortisol is secreted to restore homeostasis (*Karatsoreos and McEwen, 2010*; *Romero and Beattie, 2022*). Cortisol is produced in response to physical as well as psychosocial stressors (*Kirschbaum and Hellhammer, 1994*; *McEwen, 2017*). In children, salivary cortisol levels increase during traumatic family events and/or in anticipation of important positive or negative events, indicating that cortisol measurements are a valuable tool to assess children's stress responses to family and social interactions (*Flinn et al., 2012*; *Flinn et al., 2011*). We expected that TTS is experienced by the older offspring as a challenging event. Therefore, we predicted a sudden increase in cortisol levels at the time of sibling birth in reaction to this event.

Neopterin is produced by macrophages, monocytes, and dendritic cells after activation. Therefore, an increase in neopterin levels reflects the activation of cell-mediated immune response after an infection with intracellular pathogens (*Murr et al., 2002*). Immune responses are linked to changes in cortisol levels. While a short increase in cortisol levels can support immune functions, long-term elevation of cortisol levels suppresses immune function (*Dhabhar, 2014*). Therefore, if TTS stimulates a short increase in cortisol levels, we expect increasing or unchanged neopterin levels, whereas if TTS causes a long-lasting cortisol response, we expect a decline in neopterin levels.

Triiodothyronine (T3) is a thyroid hormone that influences metabolic rate. T3 levels decline during times of energy restriction to conserve energy (reviewed in *Behringer et al., 2018*). Measuring total T3 levels allows for disentangling the effects of energetic and social stressors, which may both occur

**Table 1.** Summary of the main findings of analyses of physiological markers and scores of older offspring behavioral during the transition to siblinghood (TTS).

| | Cortisol | Neopterin | Total T3 | Nursing | Riding | 5m-proximity with mother | Body contact with mother | Independent foraging |
|---|---|---|---|---|---|---|---|---|
| Sudden change at sibling birth | Yes, increase | Yes, decrease | No | No | Yes | No | Yes, but increase | No |
| Effect of TTS attenuates with offspring age | No | No | No | No: all changes before sibling birth | Yes, effect exists only up to 5 years old | No | No | No: all changes finished before sibling birth |

around the age of nutritional weaning and/or TTS (*Maestripieri, 2018*; *Mandalaywala et al., 2014*). If sibling birth induces metabolic issues in the older offspring, we would expect a decline in total T3 levels following sibling birth.

We complemented physiological measures with behavioral scores of nipple contact and riding, the time offspring spent in body contact with their mothers, 5 m proximity to the mother, and independent foraging. Changes in these parameters can indicate nutritional weaning and attainment of physical and social independence around TTS. We compared measures of these parameters in the older offspring before versus after the birth of a sibling to investigate whether TTS-related changes in cortisol can be linked to similar changes in behavior and thus to weaning patterns and changes in the mother–offspring relationship.

## Results

Our main results are summarized in *Table 1*, and model structures can be derived from *Tables 2 and 3*, and from *Supplementary file 1*. We applied nonlinear generalized additive mixed models (GAMM) to investigate continuous changes in our parameters of interest around the time of sibling birth and compared those models to identical ones in which we added a categorical distinction between before and after sibling birth to allow noncontinuous, sudden changes at sibling birth. We considered that our response variables may naturally change with offspring age. Age-related changes in our response variables might (a) directly mediate potential changes during TTS in case of strong temporal overlap and (b) moderate these effects as the impact of TTS may decline with decreasing dependency of older offspring from maternal support. To control for potential mediation, we ran a model with age for all our response variables. If TTS has effects beyond weaning, we would expect sudden changes at the time of sibling birth also after controlling for age-related changes. To investigate whether continuous and sudden effects of sibling birth decrease with increasing age at sibling birth, we split individuals along the median (5.11 years old at sibling birth) and ran additional models that allowed for different trajectories around sibling birth for the two age cohorts. Finally, we generated continuous two-way interaction plots to visually inspect whether and how the trajectories around sibling birth changed with increasing offspring age (for more details see 'Methods').

### Physiological changes during TTS
#### Urinary cortisol level changes in response to TTS
At the time of sibling birth, older offspring's cortisol levels showed a significant and sudden, noncontinuous, up to fivefold increase from the level prior to this event (cortisol model with one sudden change; *Figure 1A–C*, *Table 2*). Compared to a model that allowed for nonlinear, but only continuous, fitting of the data (cortisol model without sudden change, *Figure 1—figure supplement 1A and B*), allowing for discontinuity (i.e., a sudden change) in cortisol levels at the time of sibling birth (cortisol model with sudden change), significantly improved model fit (*Figure 1—figure supplement 1A and B*; $Chi^2(1) = 9.30$, $p<0.001$), even if the continuous model was allowed to be wiggly and to overfit the data (*Figure 1—figure supplement 2A*).

Post-hoc visual inspection of urinary cortisol levels indicated that urinary cortisol remained high for a long time. None of the older offspring's samples collected during the months after sibling birth had low cortisol levels (*Figure 1A–C*, *Figure 1—figure supplement 3*); cortisol measures in all samples collected within 7 months following sibling birth were above the upper 99.9% confidence interval of

**Table 2.** General additive mixed model results for physiological changes (urinary cortisol, urinary neopterin, and urinary total T3 levels; all log-transformed) in the older offspring 7 years before and after sibling birth.

| Factor variables: | Reference Category | Log cortisol | | | | Log neopterin | | | | Log total T3 | | | |
|---|---|---|---|---|---|---|---|---|---|---|---|---|---|
| | | Est. | SE | t | p | Est. | SE | t | p | Est. | SE | t | p |
| (intercept) | | 0.85 | 0.05 | 16.23 | | 2.41 | 0.04 | 58.68 | | 0.89 | 0.05 | 17.26 | |
| Males | Females | 0.11 | 0.05 | 2.09 | 0.037 | –0.03 | 0.04 | –0.69 | 0.488 | –0.11 | 0.05 | –2.19 | 0.030 |
| After S-birth* | Before† | 0.43 | 0.08 | 5.27 | <0.001 | –0.19 | 0.06 | –3.01 | 0.002 | 0.13 | 0.08 | 1.61 | 0.114 |
| Smooth term variables: | | edf | Ref. df | F | p | edf | Ref. df | F | p | edf | Ref. df | F | p |
| Time-S-birth: males | | 2.65 | 3.17 | 1.17 | 0.262 | 1.00 | 1.00 | 0.53 | 0.469 | 1.00 | 1.00 | 3.76 | 0.054 |
| Time-S-birth: females | | 1.77 | 2.12 | 0.69 | 0.433 | 1.50 | 1.83 | 0.56 | 0.603 | 1.00 | 1.00 | 0.01 | 0.922 |
| Time-S-birth: males | Females | 1.00 | 1.00 | 0.05 | 0.818 | 1.00 | 1.00 | 0.23 | 0.585 | 1.00 | 1.00 | 2.85 | 0.093 |
| Age: males | | 1.00 | 1.00 | 3.22 | 0.074 | 3.19 | 3.74 | 4.25 | 0.002 | - | - | - | - |
| Age: females | | 1.00 | 1.00 | 1.37 | 0.243 | 1.00 | 1.00 | 0.03 | 0.874 | - | - | - | - |
| Age: males | Females | 1.00 | 1.00 | 0.43 | 0.513 | 2.17 | 2.63 | 1.08 | 0.243 | - | - | - | - |
| Daytime | | 1.20 | 1.37 | 29.27 | <0.001 | 1.00 | 1.00 | 4.82 | 0.029 | 2.10 | 2.56 | 1.77 | 0.142 |
| Seasonal effect | | 2.38 | 3.00 | 11.18 | <0.001 | 0.51 | 3.00 | 0.22 | 0.278 | 0.00 | 3.00 | 0.00 | 0.723 |
| *Random effects:* | | | | | | | | | | | | | |
| Time-S-birth per ID (smooth) | | 0.00 | 111.0 | 0 | 0.238 | 0.00 | 112.0 | 0 | 0.850 | 7 | 148.0 | 0 | 0.015 |
| Age per ID (smooth) | | 0.00 | 109.0 | 0 | 0.291 | 0.00 | 108.0 | 0 | 0.737 | - | - | - | - |
| Mother ID (intercept) | | 0.00 | 13.0 | 0 | 0.175 | 0.00 | 13.0 | 0 | 0.313 | 0 | 13.0 | 0 | 0.230 |
| Year (intercept) | | 0.00 | 1.0 | 0 | 0.012 | 0.00 | 1.0 | 0 | 0.277 | 0 | 1.0 | 0 | 0.850 |
| $R^2_{adj}$ (deviance explained) | | 0.311 (33.8%) | | | | 0.169 (19.6%) | | | | 0.117 (15.3%) | | | |
| N (p-value, full/null comp) | | 319 (<0.001) | | | | 314 (<0.001) | | | | 319 (0.020) | | | |

Green indicates classic interaction term derived from a separate model calculation (see 'Methods'). Data points are physiological measures corrected for specific gravity (SG). All smooths are not controlled for age to show cumulative pattern.

ID = individual; T3 = total triiodothyronine; S-birth = sibling birth. .

*After sibling birth.

†Before sibling birth.

the values from before sibling birth. Lower cortisol values appeared later, only after 7 months post-birth (*Figure 1—figure supplement 3*). To verify this unexpected pattern, we ran another model allowing for an additional discontinuity in cortisol levels, that is, one at sibling birth and another one 7 months later. This model (*Supplementary file 1*) significantly improved model fit (cortisol model with two sudden changes compared to the model with only one sudden change at sibling birth, *Figure 1—figure supplement 4A*: $Chi^2(1) = 18.36$, p<0.001; compared with the continuous model without sudden change, *Figure 1—figure supplement 1A*: $Chi^2(2) = 27.65$, p<0.001). Cortisol levels in samples collected after the 7-month period were not different from before sibling birth (*Supplementary file 1*). While the model with the two discontinuities describes our data better mathematically, there is no obvious biological explanation for the second change (i.e., the sudden decline in cortisol) after 7 months. However, also in the model with only one discontinuity at sibling birth and a smooth continuous decline thereafter (*Figure 1A–C*), the cortisol levels took over 7 months to return to previous levels. Hence, the absence of low cortisol levels after sibling birth was evident in both models.

**Table 3.** Generalized additive mixed model (GAMM) results of behavioral changes (nipple contact, riding, and body contact and 5 m proximity with the mother) in the older offspring around sibling birth (±2 years).
Binomial GAMMs on proportions of time per day and individual.

| Factor variables: | Reference Category | Nipple contact Est. | SE | z | p | Riding Est. | SE | z | p | Proximity Est. | SE | z | p | Body contact with mother Est. | SE | z | p |
|---|---|---|---|---|---|---|---|---|---|---|---|---|---|---|---|---|---|
| (intercept) | | -7.34 | 0.84 | -8.73 | | -1.36 | 0.52 | -2.65 | 0.001 | 0.21 | 0.16 | 1.30 | 0.51 | -2.79 | 0.19 | -14.50 | |
| | Males | 0.76 | 0.61 | 1.24 | 0.21 | 1.18 | 0.36 | 3.22 | 0.001 | -0.08 | 0.13 | -0.66 | 0.51 | 0.22 | 0.11 | 2.05 | 0.040 |
| After YS-birth* | Before† | 1.39 | 0.88 | 1.55 | 0.12 | -2.00 | 0.55 | -3.64 | <0.001 | 0.02 | 0.10 | 0.18 | 0.85 | 0.47 | 0.11 | 4.17 | <0.001 |
| Year | | - | - | - | - | - | - | - | - | - | - | - | - | - | - | - | - |

| Smooth term Variables: | | Nipple contact edf | Ref. df | Chi² | p | Riding edf | Ref. df | Chi² | p | Proximity edf | Ref. df | Chi² | p | Body contact with mother edf | Ref. df | Chi² | p |
|---|---|---|---|---|---|---|---|---|---|---|---|---|---|---|---|---|---|
| T-S-birth: males | | 4.45 | 4.95 | 12.38 | 0.002 | 1.00 | 1.00 | 0.03 | 0.86 | 1.00 | 1.00 | 6.91 | 0.009 | 1.00 | 1.00 | 12.48 | <0.001 |
| T-S-birth: females | | 3.72 | 4.19 | 12.38 | 0.017 | 1.00 | 1.00 | 5.24 | 0.022 | 3.28 | 3.37 | 8.03 | 0.032 | 1.00 | 1.00 | 28.70 | <0.001 |
| T-S-birth: males | Females | 1.00 | 1.00 | 0.20 | 0.66 | 1.00 | 1.00 | 2.78 | 0.095 | 1.00 | 1.00 | 0.05 | 0.83 | 1.00 | 1.00 | 1.04 | 0.31 |
| Age: males | | 1.00 | 1.00 | 0.55 | 0.46 | 1.00 | 1.00 | 19.89 | <0.001 | - | - | - | - | - | - | - | - |
| Age: females | | 1.00 | 1.00 | 0.12 | 0.71 | 1.00 | 1.00 | 4.39 | 0.036 | - | - | - | - | - | - | - | - |
| Age: males | Females | 1.00 | 1.00 | 0.00 | 0.99 | 1.00 | 1.00 | 4.38 | 0.036 | - | - | - | - | - | - | - | - |
| Daytime | | 3.64 | 3.92 | 14.31 | 0.012 | 3.51 | 3.85 | 7.46 | 0.09 | 3.81 | 3.98 | 170.67 | <0.001 | 3.96 | 4.00 | 456.4 | <0.001 |
| Seasonal effect | | 1.10 | 3.00 | 1.89 | 0.021 | 8.89 | 3.00 | 1.91 | 0.063 | 0.00 | 3.00 | 0.00 | 0.05 | 2.64 | 3.00 | 61.22 | <0.001 |
| *Random effects:* | | | | | | | | | | | | | | | | | |
| Time-S-birth per ID (smooth) | | 3.13 | 113.00 | 82.07 | <0.001 | 33.32 | 70.00 | 251.9 | <0.001 | 62.11 | 76.00 | 1323.3 | <0.001 | 58 | 76.0 | 1569 | <0.001 |
| Age per ID (smooth) | | 8.05 | 94.00 | 34.07 | <0.001 | 0.00 | 61.00 | 0.00 | 0.010 | - | - | - | - | - | - | - | - |
| Mother ID (intercept) | | 2.45 | 10.00 | 0.00 | <0.001 | 0.00 | 10.00 | 0.00 | 0.001 | 6.28 | 12.00 | 0.00 | <0.001 | 0.01 | 12.0 | 0.01 | <0.001 |
| Date (intercept) | | 3.65 | 1.00 | 0.00 | 0.23 | 0.00 | 1.00 | 0.00 | 0.28 | 0.00 | 1.00 | 0.00 | <0.001 | 0.00 | 1.00 | 0.00 | <0.001 |
| $R^2_{adj}$ (deviance explained) | | 0.39 (62.4%) | | | | 0.827 (81.7%) | | | | 0.226 (29.3%) | | | | 0.319 (39.7%) | | | |
| N (p-value, full/null comp) | | 545 (<0.001) | | | | 301 (<0.001) | | | | 545 (<0.001) | | | | 545 (<0.001) | | | |

Green: classic interaction term derived from a separate model calculation (see 'Methods'). Statistics for year (categorical control variable) not shown for clarity.
ID: individual; S-birth = sibling birth; ':' = interaction term.
*After sibling birth.
†Before sibling birth.

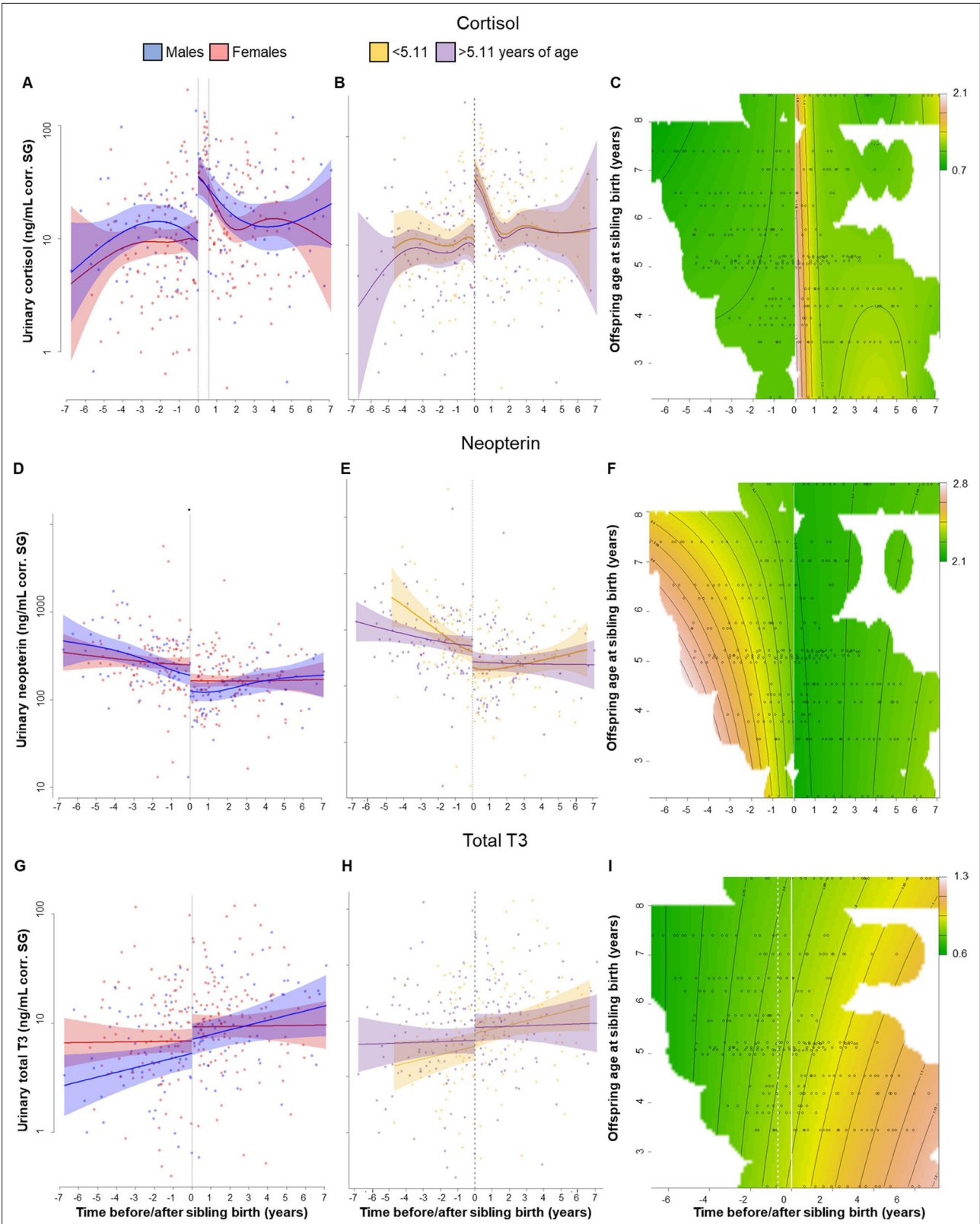

**Figure 1.** Physiological changes in cortisol (**A–C**), neopterin (**D–F**), and total T3 (triiodothyronine) (**G–I**) levels in the older offspring, 7 years before and after sibling birth (sibling birth at 0). Data points are physiological measures corrected for specific gravity (SG). All smooths are not controlled for age to show cumulative pattern. Axes for physiological variables are log-transformed. 95% confidence intervals are plotted. Left-hand plots (**A, D, G**): sex-specific trajectories around sibling birth (blue: males; red: females). Middle plots (**B, E, H**): age-specific trajectories around sibling birth for offspring

*Figure 1 continued on next page*

*Figure 1 continued*

that were older (purple) or younger (yellow) than the median value of 5.1 years at sibling birth. Right-hand plots (**C, F, I**): interaction plots visualizing how trajectories around sibling birth change with increasing offspring age at sibling birth (scale from dark green [lowest levels] to brown [highest levels]; white space: extrapolation would be unreliable due to lacking data) for the respective perspective plots, see *Figure 1—figure supplement 5*. (**A**) Urinary cortisol levels showed a significant, sudden rise to fivefold values at sibling birth (dotted line); no sex differences or age effects. (**B, C**) The sudden rise in cortisol levels was independent of the age of the older offspring at sibling birth. (**D**) Urinary neopterin levels decreased by 1/3 at sibling birth (dotted line; no sex differences or age effects). (**E, F**) The sudden decrease in neopterin levels was independent of the age of the older offspring at sibling birth. (**G–I**). Urinary total T3 levels increased around sibling birth, but this effect was indistinguishable from a general age effect. There was no significant sudden change at sibling birth in total T3 levels (**G**), and there was no significant effect of the age at sibling birth (**H, I**).

The online version of this article includes the following figure supplement(s) for figure 1:

**Figure supplement 1.** Explanation of the principal concept of the applied series of statistical models using the example of our main analysis on offspring urinary cortisol and neopterin levels 7 years before and after sibling birth (0 marks the time of sibling birth).

**Figure supplement 2.** Different models of continuous smooths of (**A**) cortisol and (**B**) neopterin levels around sibling birth that are allowed for high levels of wiggliness and thus overfitting.

**Figure supplement 3.** Scatter plot of the older sibling urinary cortisol data (blue: males; red: females) in relation to sibling birth (**A**) with a vertical dotted line at sibling birth (sibling birth is at 0) and (**B**) with two vertical dotted lines, one at sibling birth and the second one at the end of a 7-month period.

**Figure supplement 4.** Physiological changes in cortisol (**A–C**) and neopterin (**D–F**) levels in the older offspring 7 years before and after sibling birth (sibling birth is at 0) with a sudden change at sibling birth and a second sudden change after a 7-month period (cortisol) or 4.5-month period (neopterin).

**Figure supplement 5.** Perspective plots of cortisol, neopterin and total T3, showing how the trajectories of the physiological measures change with increasing age of the offspring at sibling birth.

Cortisol trajectories around sibling birth were independent of the age of the older sibling. Allowing for different levels and trajectories in older and younger individuals did not improve the model (*Figure 1B*, *Figure 1—figure supplement 4B*; $Chi^2(3) = 0.28$, p=0.91), suggesting that the cortisol level changes were not moderated by offspring age, a finding that was also apparent from visual inspection of continuous interaction plots (see *Figure 1C*, *Figure 1—figure supplement 5A, B* for perspective plots). Hence, the effect of TTS did not decrease with increasing age of the older sibling. Introducing two sudden changes, cortisol trajectories around sibling birth did not decrease with increasing age of the older sibling (*Figure 1B*, *Figure 1—figure supplement 4B, C*; $Chi^2(3) = 1.12$, p=0.53) and sex of older sibling did not affect the results (*Figure 1A*, *Table 2*).

## Urinary neopterin level changes in response to TTS

Just after sibling birth, urinary neopterin levels of older offspring decreased significantly and discontinuously (neopterin model with one sudden change, *Figure 1D–F*, *Table 2*). Compared to the model allowing for nonlinear but continuous fitting of the data (neopterin model without sudden changes, *Figure 1—figure supplement 1D*), a model with discontinuity in neopterin levels at the time of sibling birth significantly increased model fit (neopterin model with one sudden change, *Figure 1D*; $Chi^2(1) = 4.28$, p=0.003), even if the continuous model allowed for extreme wiggliness and overfitting of the data (*Figure 1—figure supplement 2B*). Post-hoc visual inspection of neopterin data suggested a 4.5-month post-birth period with particularly low neopterin levels (all values during the 4.5-month post-birth period were below the mean from before or after sibling birth). Running an additional model, allowing a second discontinuity in neopterin levels at 4.5 months, slightly improved model fit (neopterin model with two sudden changes, *Figure 1—figure supplement 4D–F*; $Chi^2(1) = 1.95$, p=0.048). However, even when allowing for a second discontinuous change, neopterin levels in samples collected after sibling birth remained significantly lower than before sibling birth (*Supplementary file 1*).

Model fit did not improve when we allowed moderation of this effect by the age of the older offspring at sibling birth (allowing for different pattern in older and younger individuals: $Chi^2(3) = 1.19$, p=0.50, *Figure 1E, F*, *Figure 1—figure supplement 4E, F*), and again, there was no sex difference in neopterin levels before or after sibling birth (*Figure 1D*, *Table 2*).

## Total T3 levels during TTS

Urinary total T3 levels increased around the time of sibling birth (*Figure 1G–I*), but this change could neither be attributed to the age of the older siblings nor to the event of sibling birth. The model

including both variables was not significantly different from the null model (p=0.096). A reduced model including only the event of sibling birth but not age was significantly better than the null model (p=0.020, *Figure 1G*, *Table 2*). There was neither a significant sex effect on urinary total T3 levels during TTS nor a significant and sudden change in total T3 levels at sibling's birth (*Figure 1G*, *Table 2*; allowing for sudden change: Chi$^2$(1) = 1.27, p=0.11). Adding interaction terms with age of the older offspring did not improve the model, nor did it if allowed for differences between older and younger individuals (Chi$^2$(3) = 0.40, p=0.85; *Figure 1I*).

## Behavioral changes during TTS

### Nipple contact during TTS

The proportion of time the older offspring was observed in nipple contact showed a continuous decrease prior to sibling birth in both males and females, and reached zero about 2 months before sibling birth (*Figure 2A–C*, *Table 3*). Consequently, there was no sudden change at sibling birth in terms of nipple contact (*Figure 2A–C*, *Table 3*; Chi$^2$(1) = 0.81, p=0.20). Allowing for different trajectories depending on the age categories of the older offspring at sibling birth (younger or older than 5.11 years old at sibling birth) significantly improved the model (allowing for different pattern in older and younger individuals: Chi$^2$(3) = 4.99, p=0.019) and visual inspection of the data indicated that nipple contact persisted mainly in younger offspring (*Figure 2B and C*, *Figure 2—figure supplement 1A*).

### Riding on the mother during TTS

The proportion of time the older offspring was riding on the mother during travel continuously decreased before sibling birth, then showed a significant and sudden decline at the time of sibling birth, and remained low thereafter (*Figure 2D–F*, *Table 3*; allowing for discontinuity at sibling birth: Chi$^2$(1) = 6.06, p<0.001). Overall, sons spent significantly more time riding on their mothers than daughters, and the continuous decline before sibling birth was only significant in daughters, whereas the sudden decline at sibling birth appeared to be stronger in sons (*Figure 2D*, *Table 3*).

Adding the older offspring's age categories significantly improved the model (allowing for different trajectories in younger and older individuals: Chi$^2$(3) = 9.32, p=0.001; *Figure 2E*). Visual inspection of the data showed that the sudden decline in riding at sibling birth was only evident in older siblings belonging to the younger age cohort (less than 5.11 years old at sibling birth), whereas older siblings in the older age cohort were completely independent from maternal carrying before sibling birth (*Figure 2E and F*, *Figure 2—figure supplement 1B*). Hence, the effect of TTS on riding disappeared with increasing age of the older sibling.

### Independent foraging during TTS

There was no effect of TTS on the proportion of time that offspring spent foraging on their own at times when mothers were foraging, and none of the full or reduced models was significantly different from the corresponding null models. Visual inspection of model results revealed that the proportion of time spent foraging independently reached high levels before sibling birth and did not change during the time window around sibling birth that was considered in our models (*Figure 2—figure supplement 1A–D*). In fact, all subjects were rather independent in terms of foraging at the time of sibling birth, irrespective of their age. In particular, there was no significant discontinuity at sibling birth (*Figure 2—figure supplement 2A–D*).

### Body contact and 5 m proximity with the mother during TTS

The proportion of time that older offspring spent in body contact with or in proximity (within 5 m) to their mothers showed similar trajectories relative to sibling birth (*Figure 2G–L*). Both variables decreased before and around the time of sibling birth, reaching low levels at the time of gestation (*Figure 2G–L*, *Table 3*). This pattern could be attributed neither to the age of older offspring nor to the event of sibling birth. The models including both age and time around sibling birth were not significantly different from corresponding null models (body contact: p=0.055, 5 m proximity: p=0.062) but models without age were significantly different from the respective null models (both p<0.001, *Table 3*).

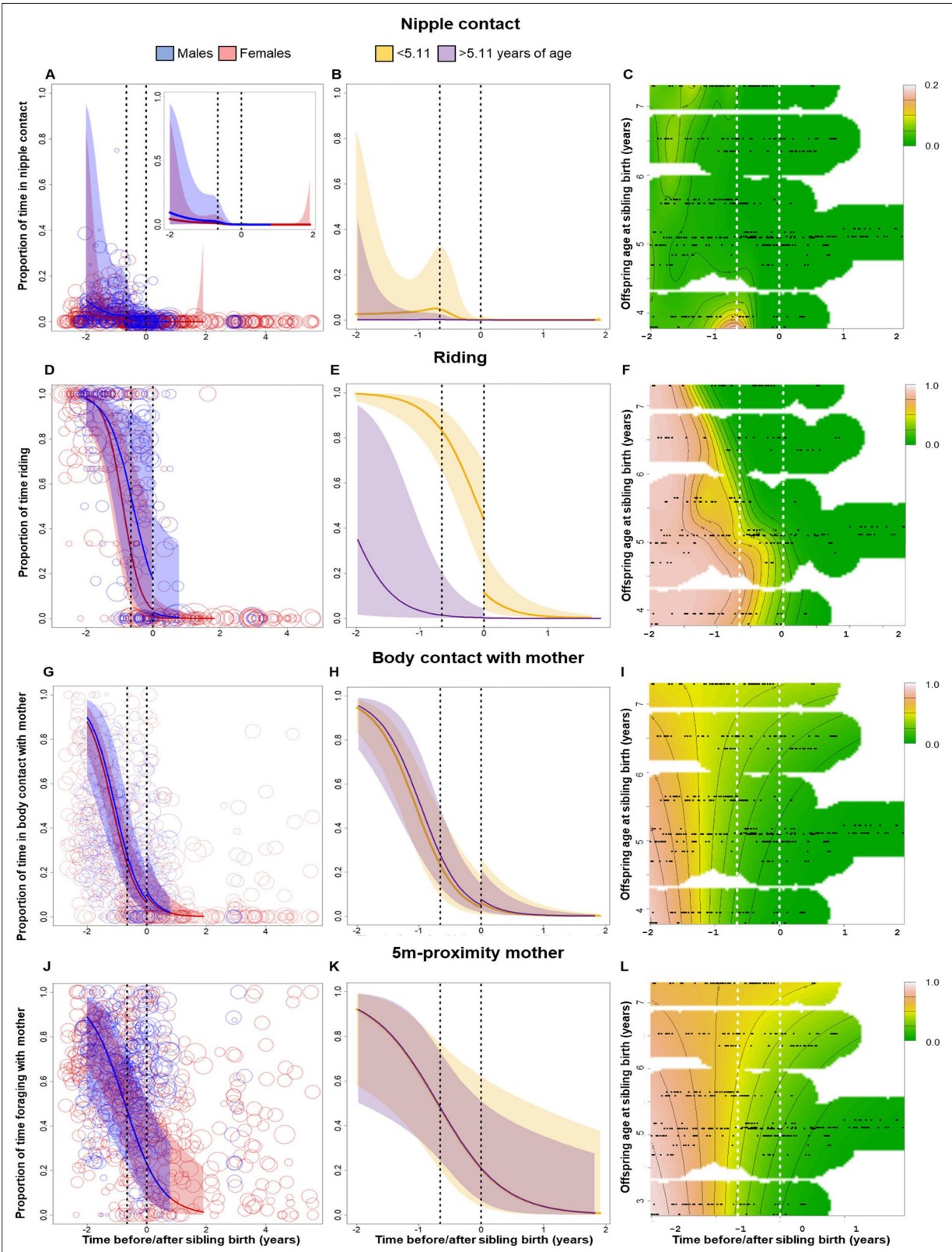

**Figure 2.** Behavioral changes in nipple contact (**A–C**), riding (**D–F**), body contact (**G–I**) and 5 m proximity (**J–L**) with the mother of the older sibling in relation to sibling birth (sibling birth is set to 0). Vertical dotted lines = time of putative conception (left dotted line) and sibling birth (right dotted line). Data points represent the proportion of time and circle size the underlying sample size (square-rooted; ranges: riding 3–44, all other behaviors 3–303). All smooths are not controlled for age to show cumulative pattern. 95% confidence intervals are plotted. Left-hand plots (**A, D, G, J**): sex-specific

*Figure 2 continued on next page*

*Figure 2 continued*

trajectories around sibling birth (blue: males; red: females). Middle plots (**B, E, H, K**): age-specific trajectories around sibling birth for offspring that were older (purple) or younger (yellow) than the median value of 5.1 years at sibling birth. Right-hand plots (**C, F, I, L**): interaction plots visualizing how trajectories around sibling birth change with increasing offspring age at sibling birth (scale from dark green [lowest levels] to brown [highest levels]; white space: extrapolation would be unreliable due to lacking data) for the respective perspective plots, see *Figure 2—figure supplement 1*. (**A–C**) Proportion of time spent suckling decreased to zero already before sibling birth (**A**) and was largely absent in older offspring (**B, C**), without a sudden change at sibling birth. (**D–F**) The proportion of time riding on the mother showed a significant and sudden decline at sibling birth (**D**), but this cut was evident only in offspring younger than 5 years old at sibling birth and not anymore in older offspring (**E, F**). (**G–I**) The proportion of time spent in body contact with the mother showed a significant and sudden increase at sibling birth, irrespective of the sex or age of the offspring. (**J–L**) The proportion of time in 5 m proximity to the mother decreased around sibling birth, but this effect was indiscernible from a general age effect. There was no significant sudden change at sibling birth (**J**), and there was no significant effect of offspring age at sibling birth (**K, L**).

The online version of this article includes the following figure supplement(s) for figure 2:

**Figure supplement 1.** Perspective plots of niplle contact, riding on the mother, body contact and proximity to the mother, showing how the trajectories of the behavioral measures change with increasing age of the offspring at sibling birth.

**Figure supplement 2.** Behavioral changes in the proportion of time spent foraging independently while the mother is foraging (to control for foraging opportunity).

For 5 m proximity, there was no sudden change at sibling birth (*Figure 2J–L*; allowing for discontinuity at sibling birth: $Chi^2(1) = 0.016$, p=0.86), nor did the pattern change with the age categories of the older sibling at sibling birth (allowing for different trajectories in younger and older individuals: $Chi^2(3) = 0.33$, p=1; *Figure 2K*).

For body contact, there was a significant sudden change at sibling birth ($Chi^2(1) = 7.60$, p<0.001), but in contrast to what one would expect to see in case of social weaning, this change was a sudden increase in body contact with the mother (*Figure 2G*, *Table 3*). Allowing moderation of the TTS effect by the age of the older offspring did not improve the model (allowing for different levels and trajectories in younger and older individuals: $Chi^2(3) = 1.86$, p=0.29; *Figure 2H and I*), and the sudden increase in body contact at the time of sibling birth was independent of the age of the older offspring at sibling birth.

## Discussion

Our data from wild bonobos demonstrate that the birth of a sibling induced a sudden increase in urinary cortisol levels in the older offspring, a physiological response that occurred in all subjects regardless of their age. Upon birth of a sibling, urinary cortisol levels in the older offspring increased fivefold and remained at this level for about 7 months. Simultaneously, neopterin levels declined at the time of the birth of a sibling and remained at low levels for about 5 months. This suggests that the birth of a sibling induced a cortisol response and reduced or suppressed cell-mediated immunity in the older offspring. Older offsrping's physiological changes around sibling birth did not decrease with increasing age of the older sibling and were independent of behavioral measures of weaning and attainment of physical independence. At sibling birth, weaning-related behavioral changes were either already completed (independent foraging and nipple contact), did not change discontinuously (urinary total T3, nipple contact, time in spatial proximity to mother, and independent foraging), changed suddenly in directions opposite of our expectation (increasing body contact time with the mother), or were significant only in subjects belonging to the younger age cohort (riding).

The fivefold increase in cortisol levels in our study is an unusually strong physiological response. For comparison, captive bonobos exposed to an experimental stress test exhibited a twofold increase in cortisol levels (*Verspeek et al., 2021*). A similar cortisol response occurred in bonobos in response to a group member giving birth, but in this case, the individual's cortisol levels returned to previous values within 1 day (*Behringer et al., 2009*). In wild chimpanzees, urinary cortisol levels were found to increase by a factor of 1.5 when subjects encounter a neighboring group, an event that exposes all group members to potentially lethal aggression (*Samuni et al., 2019*). Changes in cortisol that exceeded the magnitude of the changes observed in our study occurred in a population of wild chimpanzees who experienced a tenfold cortisol increase during a respiratory disease, which killed a number of group members (*Behringer et al., 2020*). The intensity of a stress response is generally determined by the severity, controllability, and predictability of the stressor (*Seiler et al., 2020*); TTS

is novel, severe, uncontrollable, and relatively unpredictable for the older offspring, all characteristics that likely contributed to the comparably high cortisol response that we observed in our study.

In addition to the age-independent, sudden, and substantial physiological response that we observed, a post-hoc analysis revealed that cortisol levels remained elevated for 7 months after sibling birth. Anecdotal reports indicate that, in wild chimpanzees, it may take up to 1 year until the older offspring adapts behaviorally to the presence of a younger sibling (*Clark, 1977*). While the physiological effects of sibling birth in human children are still unknown, behavioral data suggest that it may take up to 8 months until the older sibling adapts to the novel situation (*Oh et al., 2017*; *Stewart et al., 1987*). This indicates that humans, bonobos, and chimpanzees respond similarly to the challenge deriving from the arrival of a sibling.

The sudden decline of cortisol and neopterin levels to pre-sibling birth levels after 7 and 5 months, respectively, was unexpected. It is important to note that in the model with only one sudden change the cortisol levels needed many months to decline. While this result requires explanation, it is important to differentiate what our data can show and what remains to be explored in future studies. Regarding cortisol levels, our results do show that for a period of about 7 months none of the older siblings had low or average cortisol levels, but all had values within a narrow range of extremely high levels until they returned to their typical wide distribution. However, our data resolution does not allow the exact tracking of individual cortisol trajectories and it remains unclear at which time and speed different individuals return to 'normal' levels following the 7-month period. This aspect was even more pronounced in the case of neopterin levels. After the 5-month period of almost exclusively low neopterin levels, some individuals returned to previous levels but others remained low. Therefore, the time at which individuals return to 'normal' level, and the factors determining this shift remain to be investigated in future studies aiming at higher sampling rates per individual.

Cortisol levels are known to increase in response to psychological and social stressors, like predation risk or social instability, as well as energetic and physiological events (*McEwen and Karatsoreos, 2020*). Our study indicates that the sudden and persistent increase in cortisol levels in the older sibling was not related to energetic stress. Neither urinary total T3 levels, nor nipple contact, nor time spent foraging independently from the mother showed a sudden change at the onset of TTS. Similarly, if the cortisol increase at sibling birth would have been triggered by energetic challenges, the intensity of the cortisol level change should decline with the age of the older offspring as nutritional dependency on the mother decreases with age. In our study, the age of the older offspring at sibling birth ranged from 2.3 to 8.6 years, and preliminary analyses of stable isotopes in fecal samples collected from the same population suggest that nutritional weaning terminates at the age of 4.5 years (*Oelze et al., 2020*). However, the age of the older sibling at sibling birth had no effect on the strength of the cortisol response and the behavioral changes (body contact, nipple contact, riding, and independent foraging) did not follow the sudden shift in cortisol levels at the time of sibling birth.

In conjunction, our results indicate that the sudden increase in cortisol levels is independent from nutritional weaning effects and resembles behavioral responses of human children to the birth of a sibling (*Dunn and Kendrick, 1980*; *Stewart et al., 1987*). In human children, changes at sibling birth can be age-dependent. In response to sibling birth, scores for, for example, clinging and other gestures of reassurance were negatively correlated with the age of the older sibling (*Dunn et al., 1981*; *Nadelman and Begun, 1982*; *Volling, 2012*). Thus, in children, age seems to affect the behavioral response toward, or the perception of, the arrival of a sibling. Based on the results of our study, a sibling birth event is perceived similarly and independently of age. Hence, within the scope of our behavioral metrics, cortisol patterns did not match changes in single or cumulative behavioral changes around or after sibling birth.

Sibling birth is likely to cause multiple changes in the relationship between the mother and the older offspring, and only a few of them were considered in our study. For example, cortisol levels increase in response to positive arousal in children (*Flinn et al., 2011*), and while the newborn attracts the full attention of the mother it may also attract the older sibling's interest. Accordingly, it is not possible to exclude that the response of older siblings was influenced by affiliative intentions. Mothers may not always tolerate interactions between siblings and might prevent the older one from initiating interactions, which can also result in frustration and a concomitant increase in cortisol (*Gunnar et al., 2010*; *Stroud et al., 2000*). At the time of sibling birth, the social environment of the older offspring is likely to change. For example, during the first weeks after birth, female bonobos tend to avoid

large parties and forage alone or associate with a few other females (*Douglas, 2014*). This may lead to reduced rates of interactions with similar aged immatures and increased demand for social interactions with the mother who may not always be responsive to the needs of the older offspring. Another source affecting cortisol levels is aggression from group members. In bonobos, aggression against infants is rare but juveniles of both sexes can be exposed to physical aggression from adult males. Rates of aggression were found to increase with age of the immature target and were particularly high at times when mothers of targets had given birth (*Hohmann et al., 2019*; ). Thus, when females give birth, the older offspring is likely to be exposed to multiple challenges that may affect allostatic load and require the development of coping mechanisms, an achievement that requires time.

Although body contact between the older offspring and the mother decreased with age, it also suddenly increased for a short period after sibling birth. This response is not unknown: during TTS, juvenile marmosets increase proximity to parents (*Achenbach and Snowdon, 1998*), infant rhesus macaques intensify their effort to maintain contact with their mothers (*Mandalaywala et al., 2014*), and human children exhibit increased rates of clinging behavior (*Volling et al., 2017*). In our study, we did not find consistent effects of TTS on proximity within 5 m. If such changes in proximity and body contact reflect reduced maternal attention, the older offspring may aim to regain more attention from their mothers or other caregivers (*Baydar et al., 1997*). Reduced maternal attention could contribute to the increase in cortisol levels that we found, but it is still unclear why this change persists for several months. Moreover, the most consistent effect of TTS on offspring behavior in humans was a decrease in affection and responsiveness to the mother (*Volling, 2012*), which seems to contradict this interpretation. Alternatively, young female primates are known to show a high interest in new babies (*Maestripieri and Pelka, 2002*) and the increase in body contact may reflect the interest of the older offspring in the younger sibling.

The sudden increase in cortisol and the abrupt decline in neopterin levels in our study emphasize the homeostatic challenges affecting the older offspring during TTS. It is possible that the increase in cortisol levels negatively affected cell-mediated immunity. In other mammals, stress responses to weaning had a negative effect on immunity (*Kick et al., 2012*; *Kim et al., 2011*), and stressful events were associated with changes in immune function in humans (*Herbert and Cohen, 1993*). While short-term increases in cortisol levels enhance immune functions in humans, long-lasting elevations of cortisol levels – such as those found in our study – dysregulate immune responses (*Dhabhar, 2014*). In our study, urinary cortisol and neopterin levels recovered several months after sibling birth, indicating that individuals can cope with TTS to some degree, for example, by becoming habituated to the new conditions or by recruiting social support from other group members.

Persistent early-life cortisol elevations can affect an individual's ontogeny, with long-lasting consequences for its fitness, affecting its growth trajectory, metabolism, social behavior, immunity, stress reactivity, reproduction, and life history strategies (*Berghänel et al., 2017*; *Maestripieri, 2018*; *Seiler et al., 2020*). In view of our results, such effects may contribute to the observed negative effects of sibling birth on the fitness of the older offspring in nonhuman primates (*Thompson et al., 2016*; *Tung et al., 2016*; *Zipple et al., 2019*). However, the impact of sibling birth is not necessarily that strong. For example, the presence of a sibling did not affect the hypothalamic-pituitary-adrenal axis later in life in baboons, but other early-life adversities had lasting consequences (*Rosenbaum et al., 2020*). The physiological effects caused by a normative stressor that affects most individuals, such as the birth of a sibling, should be under negative selection and would therefore be considered to be a nonadaptive trait. Alternatively, it has been suggested that early-life events of 'tolerable stress' (*McEwen and Karatsoreos, 2020*) may serve to prime subjects to develop stress resistance later in life. Moreover, TTS may accelerate acquisition of motor, social, and cognitive skills (*Azmitia and Hesser, 1993*; *Maestripieri, 2018*; *Song et al., 2016*). Siblings are not only rivals but also important social partners, and the presence of an older sibling can buffer behavioral and physiological changes in response to stressful events like TTS (*Hrdy, 2011*). Having an older sibling may enhance the development and survival of the younger sibling that contributes to the inclusive fitness of both the older sibling and the mother (*Salmon and Hehman, 2015*; *Stanton et al., 2017*). Returning to our study, future studies should integrate behavior and physiological measures to estimate the impact of TTS for the older sibling and explore the long-term effects of increasing cortisol levels. The combination of physiological and behavioral measures could help to disentangle why immature bonobos show such an intense cortisol response. This would allow testing the hypotheses

that the novel mother–infant constellation is an expression of positive valence arousal or a normative change of maturation.

To our knowledge, our study on wild bonobos is the first to investigate the physiological response during TTS and, along with other studies on nonhuman primates. In many human cultures, inter-birth intervals are shorter and children are weaned at a younger age than in wild apes (*Humphrey, 2010*; *Robson et al., 2006*), despite humans having slower development and longer ontogeny. However, parental effort varies tremendously across human cultures and is often supplemented by intense allo-maternal care (*Hrdy and Burkart, 2020*). Thus, it is possible that human children do not necessarily experience such extreme and long-lasting cortisol elevation. In some families in Western societies and traditional societies, allomaternal caregivers provide nutritional, physical, and mental support to older children (*Baydar et al., 1997*; *Kramer and Veile, 2018*), which may buffer physiological responses. However, when such social buffering systems are absent or weakly developed, as in some Western societies, older children may experience the birth of a sibling as a particularly stressful time. Studies in humans are generally biased toward middle-class families in Western industrialized countries (*Fouts and Bader, 2016*; *Volling, 2012*), and our study expands research on TTS to a nonhuman primate.

The results of our study showed that bonobos, one of humans' closest living relatives, had high cortisol levels during TTS. Together with anecdotal evidence from chimpanzees (*Clark, 1977*), the information obtained in our study may shed light on the evolutionary history of the behavioral and physiological changes associated with TTS. More detailed comparisons are required to identify the emergence of behavioral and physiological traits related to TTS, their interactions, and fitness consequences. Yet, the results obtained from wild bonobos render support to the long-standing but untested and recently questioned assumption that the birth of a sibling is a notable event for the older offspring (*Volling, 2012*; *Volling et al., 2017*). It highlights the ubiquity of this pattern across individuals and age classes, and indicates that emergence of this developmental period may not be a derived trait. Interpretation of data of nonhuman primates in an evolutionary context can lead to unjustified generalization (*Sayers et al., 2012*), and it is important to note that behavioral responses to TTS in human children are highly variable and individual- and age-dependent, ranging from aggression, emotional blackmailing, and psychological disturbances, to positive attitudes toward the new family constellation (*Volling, 2012*; *Volling et al., 2017*). This raises questions regarding the coping strategies and how they are (a) influenced by the socioecological conditions including actual parent–offspring and other caretaker relationships, (b) effective in modulating and buffering the shown physiological stress response, and (c) their phylogenetic history (*Hrdy, 2011*; *Lonsdorf et al., 2018*).

## Methods

### Study site and species

Data were collected from wild bonobos (*P. paniscus*) of the Bompusa West and East communities, at LuiKotale, Democratic Republic of the Congo. This bonobo population was never provisioned with food and lives in an intact, natural forest habitat. All subjects were habituated to human presence before the start of the study, were genotyped, and were individually known. We considered every offspring only for the next sibling birth; therefore, all older offspring in our study experienced the birth event for the first time. At the time of sibling birth, the older siblings were between 2.3 and 8.6 years old. Behavioral sampling included 397.17 hr of focal data on 11 immature females (mean = 36.11, SD = 14.70) and 253.95 hr on six immature males (mean = 42.33, SD = 27.62). Physiological measurements were performed using 319 (220 females, 99 males) urine samples of 20 females and 6 males (see *Supplementary file 2*).

### Behavioral data collection and analysis

Behavioral data were collected between July 2015 and July 2018 via focal animal sampling (*Altmann, 1974*) whereby an infant was observed for 1 hr and its instantaneous behavior recorded at 1 min intervals (a detailed description in *Lee et al., 2020*). Data points were only included when focal subjects were continuously visible throughout the focal interval. Behaviors included nipple contact, defined as the infant applying its mouth to the nipple of the mother in a suckling manner, and riding, defined as the infant being transported as it clings ventrally or dorsally to its mother. For riding, we only considered data where the mother was traveling for at least three consecutive minutes to exclude situations

where the mother was likely traveling for short distances only and riding on the mother would not have been important for the offspring. We recorded when the offspring was in body contact or within 5 m proximity to the mother and when it was foraging independently (i.e., searching for its own food instead of being food provisioned by the mother). For independent foraging of the offspring, we only considered scans where also the mother was foraging to cover typical foraging situations and reduce the influence of potential sampling bias, with foraging encompassing handling and ingesting food. For all other behaviors, all scores were considered and we calculated the proportion of instantaneous records per observation day.

## Urine sample collection and analyses

Urine samples were collected between July 2008 and August 2018. Samples were collected opportunistically throughout the day between 5 am and 6 pm capturing urine directly from leaves or pipetting urine from the vegetation. Samples that were contaminated with feces were excluded. Samples were protected from direct sunlight to avoid degradation and stored in liquid nitrogen upon arrival at camp on the same day. Samples were shipped frozen to the Max Planck Institute for Evolutionary Anthropology in Leipzig, Germany, for cortisol and total triiodothyronine analysis, and later to the German Primate Center, Göttingen, Germany, for neopterin measurement.

Our urine dataset consisted of 16.0 ± 5.6 samples per individual (mean ± SD), with on average 7.5 samples before and 8.4 samples after sibling birth. Urine samples were temporally normally distributed around the day of sibling birth. Urine samples were collected from all individuals also during the first year after sibling birth, though one male and two females did not contribute samples during the first 7 months after sibling birth, and therefore, contributed only to the estimates of the urinary cortisol levels before and after the elevated cortisol period (see 'Results').

Frozen samples were first thawed at room temperature, shaken for 10 s (VX-2500 Multi-tube Vortexer), and centrifuged for 5 min at 2.000 × *g* (Multifuge Heraeus), after which specific gravity (SG) was measured using a refractometer. All results were corrected for SG to adjust the concentration of the physiological marker for urine concentration of the specimen, which depends on an individual's hydration status and time since last urination (*Miller et al., 2004*). Aliquots of samples were prepared at this time for later neopterin and total T3 analyses. In order to exclude a methodological effect concerning the order of the samples, for example, that all post-sibling birth samples are run together, all samples were randomly assigned to the measurements.

### Urinary cortisol analyses

We extracted and measured urinary cortisol in 319 (220 females, 99 males) urine samples of 20 females and 6 males. Cortisol extraction from urine samples was performed following the protocol described in *Hauser et al., 2008* for liquid chromatography–tandem mass spectrometry (LC-MS/MS) analyses. Each urine sample was mixed with an internal standard (prednisolone, methyltestosterone, d3-testosterone, d4-estrone, and d9-progesterone). Prednisolone was used as an internal standard to assess sample recovery and quantify urinary cortisol levels. We performed hydrolysis using β-glucuronidase from *Escherichia coli* (activity: 200 U/40 µl). Extracts were purified by solid-phase extractions (Chromabond HR-X SPE cartridges: 1 ml, 30 mg), followed by a solvolysis with 2.5 ml ethyl acetate and 200 mg sulfuric acid. The extraction of cortisol was carried out with methyl *tert*-butyl ether. Finally, we reconstituted evaporated extracts in 30% acetonitrile.

For urinary cortisol measurement, we used a LC-MS/MS with a Waters Acquity UPLC separation module equipped with a binary solvent manager, sample manager, and a column oven (Waters, Milford, MA). A Waters Acquity BEH C18 column (2.1 × 100 mm, 1.7 µm particle diameter) was used for chromatographic separation. Eluent A was water with 0.1% formic acid and eluent B was acetonitrile. We injected 10 µl of sample extract. The quantitative analysis of cortisol levels was realized in the range of 0.01–100 pg/µl. For cortisol quantification, we used MassLynx (version 4.1; QuanLynx Software). Final urinary cortisol results are represented in ng/ml corrected for SG. We accepted measurements of a batch if quality control measurements deviated less than 15% from the true cortisol concentration. Seventeen samples in which internal standard recovery deviated by more than 60% of the internal standard were remeasured via reinjection. In two samples, measurements were above the limit of the calibration curve, and were reinjected at a 1:10 dilution.

## Urinary neopterin analyses

We measured urinary neopterin in 314 (215 females, 99 males) aliquots of 20 females and 6 males with a commercial neopterin ELISA for humans, previously validated to determine neopterin in bonobo urine (*Behringer et al., 2017*). Prior to neopterin measurement, urine samples were diluted (1:10–1:200 depending on SG) with the assay buffer provided by the supplier. We added to each well on the plate 20 µl of the diluted urine, 100 µl of the provided enzyme conjugate, and 50 µl of the neopterin antiserum. The plate was covered and incubated on an orbital shaker at 500 rpm in the dark for 90 min. The plate was then washed four times with 300 µl washing buffer, and 150 µl of tetramethylbenzidine substrate (TMB) solution was added. The plate was incubated again for 10 min, and the reaction was stopped by adding 150 µl of the provided stop solution. Optical density was measured photometrically at 450 nm.

All samples were measured in duplicates according to the supplier's instructions. Inter-assay variation for high- and low-value quality controls was 4.2 and 1.7% (N = 17 assays), respectively. Intra-assay variation was 8.9%. Final neopterin concentrations are expressed in ng/ml corrected for SG.

## Urinary total T3 analyses

We measured total T3 in 319 (220 females, 99 males) urine aliquots of 20 females and 6 males with a commercial, competitive total triiodothyronine (T3) ELISA (Ref. RE55251, IBL International GmbH, Hamburg, Germany). Samples were measured with a 1:2, 1:5, or without dilution depending on SG. Then, 50 µl of the diluted sample with 50 µl of the provided assay reagent was pipetted into a well. We shook the plate for 10 s and incubated the plate afterward for 30 min at room temperature. We then added 50 µl of the provided triiodothyronine-enzyme conjugate to each well, shacked the plate again for 10 s, and incubated it again at room temperature for 30 min. We then washed the plate five times with 300 µl of the washing buffer and added 100 µl of TMB substrate. After 10 min of incubation, we stopped the reaction with 100 µl of the provided stop solution and read the plate at 450 nm with a microplate reader.

All samples were also measured in duplicates. Inter-assay variation for high- and low-value quality controls was 6.3 and 5.6% (N = 25 assays), respectively. Intra-assay variation was 7.2%. Final total T3 concentrations are expressed in ng/ml corrected for SG.

## Statistical analysis

All statistical analyses were performed with R 4.1.3 (*R Core Development Team, 2020*), and all R-codes can be found in the data depository. We applied GAMM, which allow for the detection and analysis of complex nonlinear relationships (termed 'smooths') that are typical for developmental trajectories. We used function gam for all models (package mgcv; *Wood, 2017*), with smooth estimation based on penalized cubic regression splines. We checked for model assumptions and appropriate model settings using functions gam.check (package mgcv), and all models were inspected for and showed negligible autocorrelation (function acf_resid, package itsadug; *van Rij et al., 2020*) and overdispersion (functions testDispersion and testZeroInflation, package DHARMa; *Hartig, 2021*). Model comparisons were conducted using the function compareML (package itsadug). GAMM smooths were plotted using package itsadug (*van Rij et al., 2020*) with removed random effects. As typical for GAMMs, interaction terms with factor variables were calculated in two ways, first analyzing whether significant changes occur within each level of the grouping factor, and second whether the smooths of the different levels differ significantly from each other (the classic interaction term statistic) (*Wieling, 2018*; *Wood, 2017*).

Urinary physiological data (urinary cortisol, total T3, and neopterin) were normally distributed after log-transformation, and Gaussian GAMMs were applied. The GAMMs on mother–offspring relationship (nipple contact, riding, independent foraging, body contact, and proximity) were based on single minute-by-minute focal scan records that were summed to time proportion values per day and individual, hence we applied GAMMs with a binomial logit-link error structure on proportion data and the underlying number of scans per proportion value as weight-argument. The main predictor variable of all analyses was the temporal change of the respective response variable around sibling birth, allowing for potential sex differences (*Behringer et al., 2014*; *Leigh and Shea, 1996*).

Time around sibling birth was added in two ways into the model, first as a continuous smooth term across time, and second as a factor variable coding for the time before and after sibling birth, thereby

allowing for a sudden, noncontinuous and unconnected change right at sibling birth. This combination allowed us to model a discontinuity at sibling birth in response values (though not in the first derivative and thus the slope of the smooth) while at the same time avoiding the pitfalls of calculating separate smooths for before and after sibling birth. Significance of the discontinuity was estimated through model comparison.

Additionally, these models included potential mediating effects of age to control whether apparent TTS effects were in fact mere general age effects irrespective of TTS. Age and time around sibling birth were naturally 100% correlated within individuals and highly correlated within the entire datasets (range $r = 0.659$–$0.855$).

In a further step, we expanded these two terms of time around sibling birth to interaction terms incorporating offspring age at sibling birth to investigate a potential moderation effect of offspring age on the intensity and pattern of potential TTS effects. For this purpose, we ran two different models. First, we ran the above model but replaced the continuous age terms with a binomial variable differentiating between offspring that was older or younger at sibling birth than the median age at sibling birth (5.11 years old). We estimated the significance of a potential difference between these two age groups in trajectories around sibling birth by comparing this model with a model without this differentiation. Second, we ran a model including an interaction term between age at sibling birth and time around sibling birth, to show visually how trajectories in the response variable around sibling birth change with increasing age at sibling birth, and in particular whether specific pattern and discontinuity around sibling birth ceased with increasing offspring age.

All statistical GAMMs were controlled for repeated measurements per individual via (a) two random smooth effects (factor-smooth-interactions; for details, see *Wood, 2017*), one for individual changes over time relative to sibling birth and the other for individual changes with age (for those models that included age as predictor variable), and (b) a random intercept per mother since some mothers contributed multiple offspring. All GAMMs were controlled for year (as random intercept for hormonal data but as control variable for the 3 years of behavioral data), seasonal effects via a cyclic smooth term over the year, and for daytime effects via a smooth term over daytime. The binomial models on behavioral time proportion data included an additional random intercept of date to control for multiple measurements per day. Due to the structure of the interaction models combining age at and time around sibling birth into one interaction term, we did not additionally control for a general mediating age effect, but merged the random effects on age and mother ID to one random smooth term of age at sibling birth per mother ID.

In all models, the number of basis functions (k) was always set equal for all predictors and random smooths of time around sibling birth and of age. The number of basis functions was generally set to 10, but needed to be reduced to 6 in some cases for the full models including both a term for age and for time around sibling birth due to sample size (for all physiological variables and for riding). Additionally, k needed to be reduced to 6 also for all models on body contact and 5 m proximity to the mother since higher values often led to strong overfitting and uncertainty. We further tested for robustness of the estimated smooth parameters by setting the number of basis functions to the respective maximum value (for models without continuous age terms), which was k = 12 for all physiological responses, k = 15 for riding, and k = 25 for all other response variables. Patterns of smooth trajectories remained the same (also for body contact in this case), though naturally the parallel increase of k for both the predictor and the associated random smooth terms led to increasing identifiability constraints and thus increasing estimation uncertainty.

## Acknowledgements

We thank the Institut Congolais pour la Conservation de la Nature (ICCN) for granting permission to conduct fieldwork on bonobos and issuing export permits for urine samples. We are grateful to the people of Lompole for hosting the LuiKotale Bonobo Project and all assistants contributing to the long-term data collection at LuiKotale. Thanks to Róisín Murtagh and Vera Schmeling for lab assistance. We are also grateful to Alison Ashbury for editorial advice. We thank Stacy Rosenbaum, Iulia Bădescu, and two anonymous referees for helpful comments and suggestions that greatly improved this article. The study was supported by funding by the German Research Foundation (Deutsche Forschungsgemeinschaft; grant number BE 5511/4-1). Long-term data collection at LuiKotale is funded

by the Max-Planck-Society, Centre for Research and Conservation of the Royal Zoological Society of Antwerp, Federal Ministry of Education and Research (Germany), Leakey Foundation, Wenner-Gren Foundation, the George Washington University, and Bonobo Alive. Funding for laboratory analyses was provided by MPI-EVA and the German Primate Center.

## Additional information

### Funding

| Funder | Grant reference number | Author |
|---|---|---|
| Deutsche Forschungsgemeinschaft | BE 5511/4-1 | Verena Behringer |
| Max Planck Institute for Evolutionary Anthropology | Open access funding | Gottfried Hohmann |
| Max Planck Institute of Animal Behavior | Open access funding | Barbara Fruth |

The funders had no role in study design, data collection and interpretation, or the decision to submit the work for publication.

### Author contributions

Verena Behringer, Conceptualization, Resources, Data curation, Funding acquisition, Validation, Investigation, Methodology, Writing – original draft, Writing – review and editing; Andreas Berghänel, Conceptualization, Resources, Data curation, Software, Formal analysis, Visualization, Methodology, Writing – original draft, Writing – review and editing; Tobias Deschner, Resources, Supervision, Funding acquisition, Validation, Methodology, Writing – review and editing; Sean M Lee, Conceptualization, Data curation, Funding acquisition, Investigation, Methodology, Writing – review and editing; Barbara Fruth, Resources, Data curation, Supervision, Funding acquisition, Investigation, Writing – review and editing; Gottfried Hohmann, Conceptualization, Resources, Data curation, Supervision, Investigation, Writing – original draft, Writing – review and editing

### Author ORCIDs

Verena Behringer http://orcid.org/0000-0001-6338-7298
Andreas Berghänel http://orcid.org/0000-0002-3317-3392

### Ethics

All samples were collected noninvasively and with permission of the Institut Congolais pour la Conservation de la Nature (ICCN).

### Decision letter and Author response

Decision letter https://doi.org/10.7554/eLife.77227.sa1
Author response https://doi.org/10.7554/eLife.77227.sa2

## Additional files

### Supplementary files

• Supplementary file 1. General additive mixed model results for physiological changes (urinary cortisol and urinary neopterin, levels; all log-transformed) in the older offspring 7 years before and after sibling birth. Green: classic interaction term derived from a separate model calculation (see 'Methods'). ID = individual; S-birth = sibling birth; * before = before sibling birth; * early after = 7 and 4.5 months following sibling birth for cortisol and neopterin, respectively; * late after = time following early after. Data points are physiological measures corrected for specific gravity (SG). All smooths are not controlled for age to show cumulative pattern.

• Supplementary file 2. Number of individuals and samples/data points (in brackets) for each physiological marker or behavior shown for each sex in relation to sibling birth. sib birth = sibling birth; total T3 = total triiodothyronine.

• Transparent reporting form

### Data availability

Source data for statistics and figures in the paper is permanently stored at GRO (https://doi.org/10.25625/O1OD2I).

The following dataset was generated:

| Author(s) | Year | Dataset title | Dataset URL | Database and Identifier |
|---|---|---|---|---|
| Behringer V | 2022 | Replication Data for: Transition to siblinghood | https://doi.org/10.25625/O1OD2I | Göttingen Research Online, 10.25625/O1OD2I |

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
