## [Editor Report]

This article examines the behavioral and physiological responses of wild bonobos to the birth of a younger sibling. The findings contribute to our understanding of the effects of a major life history transition in a primate species that is closely related to humans. An important strength of this article is the novel use of a longitudinal dataset that incorporates both behavioral and physiological measures.

---

## [Decision Letter]

**Decision letter after peer review:**

Thank you for submitting your article "Transition to siblinghood causes substantial and long-lasting physiological stress reactions in wild bonobos" for consideration by *eLife*. Your article has been reviewed by 3 peer reviewers, and the evaluation has been overseen by a Reviewing Editor and Christian Rutz as the Senior Editor. The following individual involved in the review of your submission has agreed to reveal their identity: Iulia Badescu (Reviewer #3).

Essential revisions:

1) The link to the data and code needs to be fixed so that the reviewers can evaluate the data for themselves.

2) There are two potential options for addressing the interpretation issues raised by reviewer #2. The authors may either supplement their analyses with additional behavioral data that speak more directly to stress (the preferred approach) or adjust the framing and interpretation to better suit the ambiguities of the data. If the authors choose to exercise this option, I would suggest steering away from the developmental psychology perspective the paper currently relies on, and more towards physiological mediators of life history transitions. The comments of reviewers #1 and #3 can both help to guide this since they offer related advice on framing.

3) The authors need to directly address the somewhat puzzling discontinuity in cort concentrations that occurs around 7 months post-sibling birth; what is the biological plausibility of this, and what factors might explain it? How does the non-recovery of neopterin fit with the return of cort to pre-birth baselines?

4) Please pay careful attention to the specific suggestions offered by all three reviewers. The paper would benefit greatly from additional details about the analyses and the motivations for specific components of them, to better suit a general-interest audience.

*Reviewer #1 (Recommendations for the authors):*

Introduction

Lines 51-52: What does mere behavioral adjustments mean? This requires some further explanation. I assume that it is referring to e.g. the withdrawal of maternal support, etc., but it could be interpreted in different ways.

Lines 65-67: Are there really no data available on this for non-human primates? I find it hard to believe that no one has quantified the changes in mother-older offspring relationships after the birth of a new infant. But I could certainly be wrong!

Line 86: It sounds like Schino and Troisi fulfil what I was asking for above, re: a non-human primate reference-this would probably be a good reference to add above, depending on the specifics of the findings.

Line 121: What is meant by 'general changes' in cortisol levels?

Line 125: It would be helpful to quantify what rather independent (versus highly dependent) means. Spending X% of time within X meters of mom? Is it a nursing measure? Some combination?

Line 131: This is a place in which it feels like the cart is coming before the horse. It seems odd to frame this as disentangling something which we haven't established IS entangled. This would feel appropriate if, for example, this paper were building on human studies that showed a cort increase after the birth of a sibling but did not control for changes in time near mother.

Line 132: When I hear the term stress response I think of short-term, acute changes in cort. Is it more accurate to characterize this as changes in baseline cort, given the duration of something like a transition to siblinghood? To avoid this issue entirely, you could just say 'to assess physiological stress before and during TSS.'

Paragraph on lines 132-145: This paragraph is a complicated mix of background info, theoretical justification, and methods. It would be easier to read if these were separated out. W/r/t the methods here, the T3 and neopterin measures do not feel well-justified. Due to my specific background knowledge, I understand why these measures are included in the study, but for a general interest journal, this feels like it needs more explanation. It assumes a fair amount about what the reader already knows.

Lines 158-159: I'm not sure that I am a fan of the term 'age-related weaning.' Cortisol changes with age. Cort might also change due to weaning. While obviously the specific age-related changes should be trivial in the window in which weaning is occurring, this term feels like it conflates two separate processes that could each impact cort concentrations.

Results

I personally would like to see the behavioral results come before the physiological results, since potential behavioral changes are part of what the authors are proposing might drive any physiological changes.

How is independent foraging defined? This should be clarified in the results themselves, since the methods come later.

Line 251: I would say before the sibling was conceived, instead of the conception of the mother. This could be read as the mother herself being conceived.

Generally: is there any possibility that there are 'batch effects' going on in the cort data? I.e., were all of the post-birth to 7 month samples part of the same run?

Discussion

Paragraph on lines 342-357: Potentially bolstering this argument, Rosenbaum et al. 2020 (PNAS, 117(33), 20052-20062) found that having a close-in-age sibling did not predict higher adult fecal glucocorticoid levels in the Amboseli baboons. Since they feature so heavily in these citations this seems like a good way to connect the specifics of this study to the Amboseli findings.

Line 368: I would avoid the use of the word 'modern' here. I think the point is that there is a lot of variability in levels of allomaternal care amongst living humans, and that this variation may play an important role in how stressful (or not) children find TTS to be.

Line 373: I cringe at the idea that humans are not natural…we may be an unusual animal species, but this isn't the same thing 'unnatural' (a vague and here-undefined term that means different things in different parts of the literature). It would be more accurate to say something about it contributing to the comparative literature by showing that this transition happens in a closely related species, if you want to turn the focus back to humans.

Line 380: What exactly is meant by 'constellation and co-parenting?' The sibling isn't co-parenting.

Methods

I'm still not sure how independent foraging differs from just foraging. Is there such a thing as non-independent foraging? If so, what does that look like?

Lines 437-462: Unfortunately, I am not well-versed enough in LC-MS methods to evaluate this description, so I am taking it on faith that this is all standard and reasonable. My hope is that other reviewers will be better qualified to speak to this than I am.

Line 483: I assume shanked is meant to be shookϑ.

Figures and Tables

Figure 1 in general: I think there are typos in the legend for this figure. It never refers to the right-hand panels. In general, the figure captions need to be better explained. Figures should be stand-alone, and as presented these are not.

*Reviewer #2 (Recommendations for the authors):*

Lines 135-136: This same group of researchers also found that cortisol is high leading up to the days before Christmas, when children were excitedly anticipating presents from Santa (Flinn et al., 2011; https://doi.org/10.1016/j.neubiorev.2011.01.005)-not exactly traumatic. Flinn et al. interpret these rises in both kinds of situations as "arousal to social opportunities", which might be a valuable perspective for the authors to consider in their own dataset.

Analyses: For the GAMs, it would be good to see some information on how robust the presented splines are to alternative numbers of basis functions and alternative smoothing parameters.

Relatedly, I'm a little sceptical of the approach of estimating separate splines before and after sibling birth, for all of the causal inference considerations that regression discontinuity entails. A useful companion to these 'discontinuous' splines, especially for Figure 1 and Figure S1, would be estimates of a single, continuous spline for time relative to sibling birth. It's hard to know from eyeballing it, but my guess is a jump in cortisol would still be apparent, for example, but it would be more gradual and less striking than the current figures suggest.

Line 213: Is this Chi-squared value supposed to be negative?

Discussion: My public review mentions that it is difficult to interpret the finding of increased cortisol following sibling birth. At some points, the authors appear to recognize this, as they raise a number of good arguments in the discussion about potential factors leading to a TTS-timed cortisol increase. Some, like a sibling birth coinciding with the beginning of juveniles experiencing greater male aggression, have nothing to do with TTS per se. Other arguments imply a role for beneficial behavioral interactions between the older and younger sibling. Yet, the authors appear to dismiss those concerns and conclude that the cortisol increase indeed implies stress. First, I recommend at minimum addressing why numerous alternative interpretations-some that the authors identify, and some that I have identified-should be disregarded in favor of the "cortisol = stress" conclusion. Second, as I state in the public review, having data on behavioral interactions thought to be stressful would go a long way towards solidifying the interpretative elements of this paper. I don't know whether this sort of information is available, but if it is, I recommend integrating it into a revision of this manuscript. One of the main lessons from the developmental psychology literature is that there is substantial heterogeneity in TTS adjustments, which may be attributable to family dynamics or larger ecological factors. Being able to test these kinds of theories in bonobos would significantly boost the impact of this paper.

Line 304-305: The authors claim that "findings from human children show behavioral responses to sibling birth [are] independent of their actual age". True, some show that, but others do not; many claim younger children exhibit larger behavioral disruptions (e.g. Nadelman and Begun, 1982; Volling, 2012). Some expansion on why age might matter in some aspects of adjustment, but not others, would provide useful context to the discussion.

Figure 1: The right panels of Figure 1 (B, D, and F) will be very hard to interpret for the average reader: the figure captions are sparse, the contours have labels that are so small as to be invisible, no justification/explanation is given for the extrapolation parameter (and thus it won't be clear to the vast majority of people why the figure is splotchy), etc. I understand the objective of contour plots for visualizing non-linear interactions, but as they stand the figures and/or captions need to be changed to form a self-contained explanation, as figures should. I leave edits up to the authors' discretion: they could do some combination of beef up their figure captions; present a more traditional interaction plot of linear effects (i.e., present marginal trends of -1, 0, and +1 SE on offspring age), since all the splines in Figure 1 are very close to linear; or present a different kind of plot, like a perspective plot, or a plot of predicted splines using the get_predictions function in itsadug.

Figure 2: I think the caption for supplementary Figure 2 is incorrect-there is no depiction of time relative to sibling birth in this figure, just the age of the older sibling. It appears the caption was incorrectly copied over from part of Figure 2.

Data: The DOI linked for the source data doesn't work. I would need to re-evaluate the paper after having access to the raw data, so I can verify the analytical reproducibility of the key claims.

*Reviewer #3 (Recommendations for the authors):*

Abstract:

I would not identify the birth of a sibling as a developmental transition. Perhaps would be better to say a major life transition or major transition in the early life of an individual. Or if you stick with the word development, perhaps it would help to elaborate a bit to make clear why it is a developmental transition. Maybe it applied to nutritional development so that the mother-infant nutritional relationship must end with the birth of a new sibling? Perhaps you are referring to social development? Elaborating a bit might make clearer what you mean.

Should be: "Studying the transition", "in the mother-infant relationship".

Change to "evolutionarily old"- although evolutionarily old is a bit vague. Perhaps you can narrow it down to a specific time based on comparisons with other primates or mammals. For instance, you might say that this effect was likely present in the common ancestor of all the great apes. Something more specific to your study.

Line 43: change "while still being dependent" to "while they are still dependent"

This sentence also needs a reference- also do you mean children in humans only? I guess that is what you mean but perhaps you should extend this to all primates or long-lived, slow developing mammals. You could say that in mammals with slow rates of growth and development that give birth to single offspring at a time, most offspring are exposed to the birth of a younger sibling.

Extending this to other species could make it more relevant to more researchers.

Line 45: competitor in what sense? competitor for maternal attention and resources, or for food in the environment, or for reproduction or for social partners etc. I think this needs to be more specific.

Line 49: Do you mean behaviors shown by the older sibling? this is not clear. make clearer that it is the older sibling who could show increased aggression, clinginess, and depression. Otherwise, could be perceived to be in the mother or in the new sibling.

Line 51: not sure what you mean by mere behavioral adjustments. I think the point you are making here is important, but couldn't behavioral adjustments be in response to stress. I guess you are trying to say that these are behavioral changes that could occur in response to new siblings that are not associated to stress for the older sibling. Make clearer.

Line 94-97: I do think this statement needs references to support it.

Line 98: I think this statement also needs some references and a range in inter-birth interval lengths for bonobos, including a variety of sites, since you have the ones for your site specifically, below.

Line 101: should be social dependency on the mother.

Line 106: would co-dependency on the mother be better than co-residence? Co-residence sounds like one offspring needs to leave the natal group at some point, which is not always the case. Co-residence with siblings can occur for their entire lives if they are the philopatric sex.

Line 110-111: Sexual maturation at 4 years of age is very young… Many individuals might still be nursing at 4 years old. Perhaps your definition of the onset of sexual maturation needs to be explained. Also, unclear how the onset of sexual maturation is different from the onset of menarche, and how these two milestones are different, so this should be cleared up as well.

Introduction in general:

You've made many comparisons between bonobos and humans, which is good, but I wonder if it would be good to add in data and comparisons with other apes as well (chimps, gorillas, orangutans). I know you mention chimpanzees once or twice, but I wonder if it would help to make more explicit comparisons throughout with the other great apes. I say this because eventually in your discussion, you are going to be making some preliminary conclusions about when various developmental and stress mechanisms would have evolved and saying that the timing would have been in the last common ancestor between bonobos and humans may be incomplete. Inherently we would want to hear about the chimpanzee data at least, to know if we should think about the last common ancestor with Pan, but of course we would wonder about any evidence in the other great apes. If the other great apes show similar patterns in life history, development and maternal investment that you are pointing out for humans and bonobos, then the findings of your research here could also apply to them (pending studies in these other apes). Alternatively, there might be some key differences in the life history patterns and infant development of different apes, so it would be good to know about them to later understand why you may not think that the findings extend to a certain genus.

I suggest you try to keep the tense for your article in the past tense as much as possible. As it is now, you switch quite a bit between present and past tense, and at times, you use present tense but this is a bit awkward. Past tense would be best.

Line 123: You should mention the name and place of your study population here since you are bringing it up to set up your study.

Line 125: this statement on the dependency levels of the offspring needs to be supported with a reference and it needs to be quantified in some way. Is dependency based on nipple contacts, proximity, sleeping in the same nest, being carried from place to place, all the above?

Also, unclear if this statement refers to past research or if this is something that is being done in this paper. Did someone already investigate the nutritional development of these individuals, or is the level of dependency something you will be measuring in the present study and measured relative to TTS? This needs to be made clearer.

Lines 139-140: I think this sentence needs some references. Also, you explain T3 in greater detail but not the neopterin hormone. I think a sentence specifically on neopterin and how it works or what it means can help here.

Lines 140-141: I am not quite understanding T3 can help disentangle effect of energetic stress versus social stress. Can you please make this more obvious/clearer?

Line 143: unclear why neopterin levels should decrease because we do not have enough information about neopterin.

Line 146: you say nursing here but later you define suckling as contact with the nipple. Might be better to stick with one of the two terms, especially since people are going to keep an eye out for the term nursing to see how this was defined later in the text.

Line 148-150: Are you saying that social weaning is weaning from the mother more generally, in terms of proximity and access to carrying, etc.? This is very confusing because weaning in most infant development/maternal investment literature refers specifically to nursing or feeding behavior. Until this point in the article, I was thinking you were trying to make a distinction between nutritive and non-nutritive nursing.

If you would like to talk about changes in the mother-infant relationship outside of nursing and foraging behavior, I strongly encourage you NOT to use the term weaning for that. Perhaps switch to "attainment of physical independence" or some other similar phrase since the only other measures outside of nursing behavior that you are looking at are more physical measures (so the infant being carried versus moving on its own/ infant in certain proximity to the mother). Instead of nutritional and social weaning, you could say weaning and attainment of physical independence.

Line 193: With increasing age, urinary…

Also, you say that they significantly changed in males but not females, but then you say that this change was not significant… I am confused.

Line 210-211: same issue here. you say significantly declined but then in brackets you said sex difference not significant. Which is it?

Line 75, 262 and elsewhere: You keep talking about nutritional versus social weaning (or as others have said, nutritional versus behavioral weaning), but this concept is fairly recent. The idea that there are two components to weaning, the milk-transfer and the psycho-social relationship between the mother and the infant through continued nipple contacts, regardless of whether or not milk transfer occurs, is a fairly new and innovative idea that needs to be supported with literature in your paper, and that needs to be explained a little bit somewhere in the text. For example, research on wild chimps and other primates has shown that comfort nursing, without milk transfer, can occur for years after lactation has ended. This sets up a situation where you have these two separate weaning periods: weaning from milk and weaning from nipple contact- so the nutritional versus social/behavioral weaning. Thus, the mother-infant behavioral relationship can develop separately from the mother-infant nutritional relationship, despite considerable overlap between the two. This distinction, and its importance for the infant and mother, should be explained.

Line 276-279: Are you saying that TTS is severe, uncontrollable for older offspring, and was unpredictable or moderately unpredictable? Make clearer how this statement relates to your results.

Line 347-349: change sentence to

"However, strong negative effects caused by a highly predictable and normative stressor that invariably affects most individuals, such as the birth of a sibling in apes, should be under negative selection, and would thus be expected to XX XX (explain what this would mean)."

Line 298: I would change this to

"…suggest that nutritional weaning is usually completed by around 4.5 years of age".

Lines 372 to 372, or more broadly for this whole paragraph: Similar to my comment for the abstract, the statement that your results highlight the evolutionary history of stress response and TTS is vague and almost seems like a throw away idea because it is not specific enough. Can you go a step further and talk about the possible timing of an evolutionary link between stress and TTS? Presumably, this important interaction would have appeared with the great ape transition? Or would it be with the transition between genus Pan and Homo? Use the comparisons in life history traits and infant development of humans and the other great apes to infer when this mechanism that you found could have become more important or more prominent.

Methods:

Line 397: I would like to see a table with a breakdown, by infant age and sex, of the sample sizes and numbers for the behavioral and urine data.

Lines 399 to 411: Did you exclude from the calculations time out of view, or time when the infant was ventral, but it was unclear if they were in contact with the nipple, and excluded time in a nest with the mother since you can't see the infant? If yes, say so, if not, justify why you did not.

I am not familiar with any of the urine analyses so cannot comment on these.

Please explain how the proportions of the behavioral measures were calculated somewhere in the methods.

[Editors' note: further revisions were suggested prior to acceptance, as described below.]

Thank you for resubmitting your work entitled "Transition to siblinghood causes substantial and long-lasting increase in urinary cortisol levels in wild bonobos" for further consideration by *eLife*. Your revised article has been evaluated by Christian Rutz (Senior Editor) and a Reviewing Editor.

The manuscript has been greatly improved, but there are some remaining issues that need to be addressed before it can be accepted for publication.

– First, and most importantly, I (the reviewing editor) strongly urge you to consider the comments of reviewer #2 w/r/t the plausibility of the two-intercept model. While I understand that this model provides the best fit statistic, in my opinion, fit statistics should not override biological plausibility. There is no biologically plausible explanation for why cort would suddenly drop at 7 months (or at least, none that have been relayed in this text), and basic data visualization -- i.e., the raw scatterplot of the data--does not suggest this is what is happening. I personally feel that the single-cut discontinuous spline is a good middle-ground choice (due to the suddenness of sibling birth), followed by a continuous spline, followed by the two-cut discontinuous. The last two options would certainly be worth including in the supplementary materials.

This issue is the primary reason that I am recommending a revise and resubmit, rather than an acceptance. Please note that my acceptance of the manuscript is not contingent upon making this change. I respect that authors may have different points of view surrounding the interpretation of fit statistics. However, if it is not changed, the final evaluation summary will reflect the fact that myself and one of the other reviewers disagreed with the analysis strategy. I want to be transparent about the source of the disagreement so that the evaluation summary would not come as a surprise should you choose not to change which model you prioritize in the main text.

– Please include a scatterplot of the raw data in the supplementary materials.

– Reviewer #2 is correct that though the language around 'stress' has been considerably improved, there are still places where there are ambiguities in how it is used/what it is implying. The manuscript will be stronger with these ambiguities removed. These data are quite interesting and do not need to lean on 'stress' in order to be noteworthy and important. Similarly, I agree with reviewer #3s concerns about characterizing TTS as a developmental stage. Both of these are wording issues that should be simple to resolve.

– Please follow reviewer #3s request for a thorough proofread. There are a considerable number of spelling and grammar mistakes that remain, which is distracting as a reader. They have many helpful suggestions for places where the language could use additional clarification, which again, will help strengthen the final product.

*Reviewer #2 (Recommendations for the authors):*

Most of the issues I raised have been sufficiently addressed. Most prominently, the data and code are now available, and I was able to check that the authors' results are reproducible (Side note: I recommend further commenting the code so it is clear e.g. which models correspond to which figures, and I also recommend sharing the code in a different format than a Word document-these steps would help with accessibility).

I have a few remaining concerns:

Lines 75 – 78: "Accordingly, in humans, TTS is considered to be a stressful life event or even a disruptive crisis for the older sibling even under favorable conditions, a perspective that seems to be supported by TTS-related behaviors of the older offspring such as aggression, clinginess, and depressive syndromes (reviewed in Volling, 2012; Volling et al., 2017)."

As I wrote in my initial review, this claim is an oversimplification of these papers. Volling (2012), in particular, clearly concludes that there is substantial evidence for either of two quite different perspectives: that TTS is stressful, OR that it is an occasion for ecological adjustment that does not manifest in stress. This section of the intro should be edited to better reflect the ambiguity of the background literature, and the many different ways in which "stress" might be operationalized (see also my last comment below).

Lines 360 – 361: "The sudden recovery of cortisol and neopterin levels to pre-sibling birth levels after seven and five months, respectively, is puzzling and requires explanation."

I appreciate the authors expressing some caution here, but I would go further, which raises a somewhat larger point. The authors' statement I quoted is only warranted if the "two-cut" discontinuous model is taken at face value, and I am skeptical that should be done. The scatter plot of cort values, ignoring any splines, shows an apparent increase in the period right after sibling birth. That's interesting and worth trying to understand. But there are plenty of cort values just as high right before and right after the post hoc seven-month window. The "sudden-ness" of the increase and decrease is a function of specifying different intercepts at two different points. I wrote in my last review that I favored a single continuous spline for figures. The authors have basically responded by saying that "a one-cut discontinuous spline is a better fit than continuous, and a two-cut model is better still." But we have to consider biological plausibility, not just fit statistics. Do we really think it's realistic to expect such a uniform and sudden change seven months in? If we consider plausibility, I think one can make a reasonable case for a continuous spline, or one with a discontinuity at sibling birth (which is a prominent and biologically meaningful event), but not so much for the "two-cut" model. I recommend the authors foreground more biologically plausible models, and if they really still wish to include the "two-cut" model, that it occupies a less prominent position.

Lines 488 – 490: "Yet, the results obtained from wild bonobos renders support to the long-standing but untested and recently questioned assumption that the birth of a sibling is a stressful event for the older offspring (Volling, 2012; Volling et al., 2017)".

Once again, how are we to conclude that a cortisol increase equates to an individual experiencing a "stressful event"? The authors have become more careful in their language in many parts of the manuscript, but not here. The authors might say that they are simply using "stressful" to mean "a deviation from homeostasis", but my point all along has been that the authors need to be very, very clear about operationalizations if they want to use the term "stressful" in a different manner than its widely understood everyday definition. I recommend being super explicit about the usage and definition of the term "stress" in the intro, as I mention above. I mean literally provide a clear definition, upfront, so all readers are on the same page. The brief link between cortisol and homeostatic functioning, which isn't given until p. 5, is too indirect and vague. I also recommend removing statements like the one I quoted above. The authors already state in the discussion what we can actually determine here: cortisol goes up after the birth of a sibling, and it doesn't seem to be related to their age when it happens. Whether this event is "stressful" or not (which the authors claim without contextualization) cannot be determined by this dataset.

*Reviewer #3 (Recommendations for the authors):*

I am satisfied with the changes made by the authors based on my first review.

General comment: I would suggest a thorough proof-read of the text, just to make sure there aren't any little vocab or grammar mistakes and to polish it up. I caught many of the mistakes and pointed them out in my review, but I'm sure I missed some.

Title: would be better to say "causes a substantial and long-lasting increase in…"?

Abstract:

Second sentence could be worded better, as it is hard to understand. Perhaps "In these species, the birth of a sibling marks a major transition in early life, as maternal investment is constrained, and older siblings experience a decrease in maternal support."

Following sentence could be worded better:

"In the older offspring independent of its age, with siblings' birth, urinary cortisol levels increased fivefold and remained elevated for seven months."

Maybe better would be: "Following the births of siblings, urinary cortisol levels of older offspring, independent of their age, increased fivefold and remained elevated for seven months."

Change "did not show corresponding change" to "did not change". Otherwise, it is confusing for a few reasons: first, should be changes, second, make one wonder what corresponding means. I suggest you just simplify and cut out the unnecessary "show corresponding" part.

I am not a fan of the word syndrome. Sounds like a disease or physical problem. Could you say:

"Our results suggest that bonobos and humans experience TTS in similar ways and that this…".

Also, in this sentence, I do not like the idea of TTS as a developmental stage. I mentioned this in my last review and the authors made the change, but perhaps the idea got lost in the revising and then it was added back in here. TTS is more of a life stage or a life history stage… or a life transition. Would you say that offspring whose mothers never make a sibling are underdeveloped because they were not able to undergo the developmental stage of TTS? No because TTS is not a developmental stage, like weaning is, for example. So, I would reframe the idea of TTS that does not imply that it is a stage of development.

Line 50-51: but maternal provisioning through food sharing and feeding of young non-milk foods occurs after weaning. Revise this sentence to make clearer you are talking about milk and/or nursing.

Line 64-66: This sentence should be simplified and cleaned up. It is unnecessarily complicated and wordy, right now. Also, what are you trying to say exactly? That there are negative effects associated with having to share maternal care? Be more specific.

Line 69: Maybe change mother-offspring dependency to something less extreme, since dependency almost sounds like the offspring will be completely reliant on the mother for its whole life.

Line 91: change to "to be a social comfort behavior"

Also, the Badescu reference should be 2017, as it came out in early view in 2016 but then received an official issue in 2017. It is always very confusing.

Line 107-108: the last part of this sentence may not be grammatically correct.

Line 137: Not sure what constitution means. Can you replace with a more specific term?

Line 144-145: Hmm. I guess I understand what you mean but is this something measurable, that you determine from certain variables? It seems a bit speculative to say assume that the infant would die or would survive. Perhaps you can qualify these statements by offering as examples the variables you used to determine this. For example, you likely mean nursing behavior. So, the highly dependent were still regularly nursing, for prolonged periods, whereas the independent hadn't been seen nursing X focal hours of observation. Or maybe you mean the time that infants spent in body contact. Whatever you used to identify independence, give us that information.

Line: Visual inspection of?

Line 194: like here for example, should be "were" not "was". A small error but there are several like this throughout the text so go through carefully to polish up the writing.

Line 334: Do you mean "or showed a sudden change but in the opposite direction…"?

Line 340-341: levels returned to the levels before within one day. This phrase could be worded better or constructed a bit better to make easier to understand

Line 361: Wait, why is this puzzling. You just told us in the previous sentence that it takes human children 8 months to recover after sibling birth. Then the recovery you found after seven and five months is right in line with what you expect. Why puzzling?

Line 378: Unless I missed this somewhere, it would be good to add a note about whether some of the siblings continued to make nipple contacts after the birth of a new baby. And maybe tell us how many of the total number of infants were observed to continue nursing after birth of baby. This is interesting information for others. Maybe in results would be good to add?

Line 382: Say what kind of samples. Fecal samples? Hair samples? Urine samples? Bone samples?

Line 403: Affiliative intends? Not sure what you mean here.

Line 449: In baboons'? Maybe in baboons,…? Something is off here. Also, baboons repeats a few times. Some errors need fixing

Line 486: I don't like the word syndrome in this context. Makes it sound like TTS is a disease.

In methods, might be good to mention which authors collected which data, which authors did which analyses, etc. For example, the behavioral data were collected by focal animal sampling. Was this done by all the authors together, or just one? Give the initials of the author(s).

---

## [Author Response]

Essential revisions:1) The link to the data and code needs to be fixed so that the reviewers can evaluate the data for themselves.

We apologize, we have now fixed this issue. Data as well as code are available.

2) There are two potential options for addressing the interpretation issues raised by reviewer #2. The authors may either supplement their analyses with additional behavioral data that speak more directly to stress (the preferred approach) or adjust the framing and interpretation to better suit the ambiguities of the data. If the authors choose to exercise this option, I would suggest steering away from the developmental psychology perspective the paper currently relies on, and more towards physiological mediators of life history transitions. The comments of reviewers #1 and #3 can both help to guide this since they offer related advice on framing.

We do not have behaviour data to clearly show that the increase in cortisol levels was a stressor. Throughout the manuscript we describe cortisol increase as a response to homeostatic threat. We also followed the suggestion of reviewer 1 and 3 not to label it stress response and we discuss more options, also positive homoeostatic threats.

3) The authors need to directly address the somewhat puzzling discontinuity in cort concentrations that occurs around 7 months post-sibling birth; what is the biological plausibility of this, and what factors might explain it? How does the non-recovery of neopterin fit with the return of cort to pre-birth baselines?

We have now included a whole paragraph in the discussion (line 351-363).

4) Please pay careful attention to the specific suggestions offered by all three reviewers. The paper would benefit greatly from additional details about the analyses and the motivations for specific components of them, to better suit a general-interest audience.

We are grateful for all the comments and help the reviewer provided. See the specifications below.

Reviewer #1 (Recommendations for the authors):IntroductionLines 51-52: What does mere behavioral adjustments mean? This requires some further explanation. I assume that it is referring to e.g. the withdrawal of maternal support, etc., but it could be interpreted in different ways.

We have now explained in detail- also following reviewer 3: “However, whether behavioral changes during TTS are associated with sibling birth rather than age-related behavioral adjustments related to withdrawal of maternal support remains to be resolved” (lines 79-81)

Lines 65-67: Are there really no data available on this for non-human primates? I find it hard to believe that no one has quantified the changes in mother-older offspring relationships after the birth of a new infant. But I could certainly be wrong!

We have now re-worded the introduction to show that this is not a human only topic and following also the suggestion of the editor more in the direction of a life history event (lines 50-79).

Line 86: It sounds like Schino and Troisi fulfil what I was asking for above, re: a non-human primate reference-this would probably be a good reference to add above, depending on the specifics of the findings.

See answer above.

Line 121: What is meant by 'general changes' in cortisol levels?

We have now re-worded the sentence to better explain what was shown: Notably, monitoring changes of urinary cortisol levels revealed first evidence that older offspring may respond physiologically to the birth of a sibling (lines 129-130)

Line 125: It would be helpful to quantify what rather independent (versus highly dependent) means. Spending X% of time within X meters of mom? Is it a nursing measure? Some combination?

Following also reviewer 3, we have now included two definitions for the terms and that this was used for our study: “… and the state of the older siblings at the time when mothers gave birth to another infant ranged from highly dependent (i.e., infant would die without mother) to rather independent (i.e., offspring would likely survive without mother)”. (lines 140-1452)

Line 131: This is a place in which it feels like the cart is coming before the horse. It seems odd to frame this as disentangling something which we haven't established IS entangled. This would feel appropriate if, for example, this paper were building on human studies that showed a cort increase after the birth of a sibling but did not control for changes in time near mother.

We have now re-written the whole part. This specific part does not exist anymore.

Line 132: When I hear the term stress response I think of short-term, acute changes in cort. Is it more accurate to characterize this as changes in baseline cort, given the duration of something like a transition to siblinghood? To avoid this issue entirely, you could just say 'to assess physiological stress before and during TSS.'

Following the suggestion by reviewer 2 we avoided to label it stress response and named it cortisol increase.

Paragraph on lines 132-145: This paragraph is a complicated mix of background info, theoretical justification, and methods. It would be easier to read if these were separated out. W/r/t the methods here, the T3 and neopterin measures do not feel well-justified. Due to my specific background knowledge, I understand why these measures are included in the study, but for a general interest journal, this feels like it needs more explanation. It assumes a fair amount about what the reader already knows.

We have now separated literature background from methods. We also have now added more information to explain why a marker was measured. We further structured the paragraph by cortisol, neopterin, and total T3 (lines 147-169).

Lines 158-159: I'm not sure that I am a fan of the term 'age-related weaning.' Cortisol changes with age. Cort might also change due to weaning. While obviously the specific age-related changes should be trivial in the window in which weaning is occurring, this term feels like it conflates two separate processes that could each impact cort concentrations.

We have now changed the terminology: If TTS had effects beyond weaning effects, then we expected sudden changes at sibling birth also after controlling for age-related changes. (lines 187-189)

ResultsI personally would like to see the behavioral results come before the physiological results, since potential behavioral changes are part of what the authors are proposing might drive any physiological changes.

We thank the reviewer for the suggestion. We prefer to start with cortisol followed by the other two markers, because cortisol was the focus of the study and the two others as well as the behaviors were analysed to provide information about the circumstances and physiological state of the offspring.

How is independent foraging defined? This should be clarified in the results themselves, since the methods come later.

We have now added a definition in the method section: For independent foraging of the offspring, we only considered scans where also the mother was foraging to cover typical foraging situations and reduce the influence of potential sampling bias, with foraging encompassing handling and ingesting food (lines 518-520).

We have now added in the results when the term was used for the first time: There was no effect of TTS on the proportion of time spent foraging on their own (during maternal feeding time, to ensure foraging opportunity), … (lines 283-284)

Line 251: I would say before the sibling was conceived, instead of the conception of the mother. This could be read as the mother herself being conceived.

We thank the reviewer for the wording suggestion. The results were re-written for independent foraging. (lines 283-291)

Generally: is there any possibility that there are 'batch effects' going on in the cort data? I.e., were all of the post-birth to 7 month samples part of the same run?

We have now added: In order to exclude a methodological effect concerning the order of the samples e.g., that all post sibling birth samples are run together, all samples were randomly assigned to the measurements. (lines 544-546)

DiscussionParagraph on lines 342-357: Potentially bolstering this argument, Rosenbaum et al. 2020 (PNAS, 117(33), 20052-20062) found that having a close-in-age sibling did not predict higher adult fecal glucocorticoid levels in the Amboseli baboons. Since they feature so heavily in these citations this seems like a good way to connect the specifics of this study to the Amboseli findings.

We are thankful that the reviewer mentioning this interesting reference. We have now added: However, the impact of sibling birth is not necessarily that strong. For example, in baboons’ presence of a sibling did not affect the HPA-axis later in life in baboons, but other early life adversities had lasting consequences (Rosenbaum et al., 2020). (lines 439-442)

Line 368: I would avoid the use of the word 'modern' here. I think the point is that there is a lot of variability in levels of allomaternal care amongst living humans, and that this variation may play an important role in how stressful (or not) children find TTS to be.

We agree with the reviewer and have now deleted modern. (line 469)

Line 373: I cringe at the idea that humans are not natural…we may be an unusual animal species, but this isn't the same thing 'unnatural' (a vague and here-undefined term that means different things in different parts of the literature). It would be more accurate to say something about it contributing to the comparative literature by showing that this transition happens in a closely related species, if you want to turn the focus back to humans.

We have now reworded the sentence following the reviewer’s suggestion: The results of our study showed that bonobos, one of humans closest living relatives, had high cortisol levels during TTS. (lines 474-475)

Line 380: What exactly is meant by 'constellation and co-parenting?' The sibling isn't co-parenting.

We agree with the reviewer and deleted co-parenting, ad added “family”: It is therefore important to note that, behavioral responses to TTS in human children are highly variable and individual- and age-dependent, and range from aggression, emotional blackmailing and psychological disturbances, to positive attitudes towards the new family constellation (Volling, 2012; Volling et al., 2017). (lines 484487)

MethodsI'm still not sure how independent foraging differs from just foraging. Is there such a thing as non-independent foraging? If so, what does that look like?

We have now added a clear definition the method section: … and when it was foraging independently (i.e., searching for its own food instead of being food provisioned by the mother). For independent foraging of the offspring, we only considered scans where also the mother was foraging to cover typical foraging situations and reduce the influence of potential sampling bias, with foraging encompassing handling and ingesting food. (lines 516-520).

Independent foraging is a term commonly used in animal behaviour publications:

For example:

Brown GR, Almond REA, Bergen Y van. 2004. Begging, stealing, and offering: food transfer in nonhuman primates Advances in the Study of Behavior. Elsevier. pp. 265–295. doi:10.1016/S0065-3454(04)34007-6

Burns JM, Clark CA, Richmond JP. 2004. The impact of lactation strategy on physiological development of juvenile marine mammals: implications for the transition to independent foraging. *International Congress Series*
**1275**:341–350. doi:10.1016/j.ics.2004.09.032

Jeglinski J, Werner C, Robinson P, Costa D, Trillmich F. 2012. Age, body mass and environmental variation shape the foraging ontogeny of Galapagos sea lions. *Mar Ecol Prog Ser*
**453**:279– 296. doi:10.3354/meps09649

Soulsbury CD, Iossa G, Baker PJ, Harris S. 2008. Environmental variation at the onset of independent foraging affects full-grown body mass in the red fox. *Proc R Soc B*
**275**:2411–2418.

doi:10.1098/rspb.2008.0705

Line 483: I assume shanked is meant to be shookϑ.

Absolutely, we have now changed shanked to shook (line 594)

Figures and TablesFigure 1 in general: I think there are typos in the legend for this figure. It never refers to the right-hand panels. In general, the figure captions need to be better explained. Figures should be stand-alone, and as presented these are not.

We have now added additional information and have re-written the figure descriptions. See also comments of reviewer 2.

Reviewer #2 (Recommendations for the authors):Lines 135-136: This same group of researchers also found that cortisol is high leading up to the days before Christmas, when children were excitedly anticipating presents from Santa (Flinn et al., 2011; https://doi.org/10.1016/j.neubiorev.2011.01.005)-not exactly traumatic. Flinn et al. interpret these rises in both kinds of situations as "arousal to social opportunities", which might be a valuable perspective for the authors to consider in their own dataset.

We are thankful for the reviewers thought, and unfortunately, we gave the impression that the cortisol stress reaction is something negative “stress”. We added therefore the suggested reference in the introduction and a sentence to show that the cortisol release is a reaction of stressor threatening homeostasis. (lines 147-155).

Analyses: For the GAMs, it would be good to see some information on how robust the presented splines are to alternative numbers of basis functions and alternative smoothing parameters.

We have now added a paragraph to the methods section where we discuss this in detail:

"Throughout models, the number of basis functions (k) was always set equal for all predictor and random smooths of time around sibling birth and of age. The number of basis functions was generally set to 10, but needed to be reduced to 6 in some cases for the full models including both a term for age and for time around sibling birth due to sample size (for al physiological variables and for riding). Additionally, k needed to be reduced to 6 also for all models on body contact and 5m- proximity to the mother since higher values often led to strong overfitting and uncertainty. We further tested for robustness of the estimated smooths parameters by setting the number of basis functions to the respective maximum value (for models without continuous age terms), which was k = 12 for all physiological, k = 15 for riding, and k = 25 for all other response variables. Patterns of smooth trajectories remained the same (also for body contact in this case), though naturally, the parallel increase of k for both the predictor and the associated random smooth terms led to increasing identifiability constraints and thus increasing estimation uncertainty". (lines 663-674).

In addition, we also run a series of models for our main result on discontinuities at sibling birth in cortisol and neopterin. Here, we allowed the model with the continuous smooth only for high wiggliness and thus higher potential to sufficiently fit the sudden change at sibling birth, by increasing k of the predictor smooth term to 50 and reducing k of the respective random smooth term to 6, plus introducing varying low smoothing parameters. We then compared these models with our unmodified discontinuity models and reported the results in the Results section and in supplemental figure S3.

Relatedly, I'm a little sceptical of the approach of estimating separate splines before and after sibling birth, for all of the causal inference considerations that regression discontinuity entails. A useful companion to these 'discontinuous' splines, especially for Figure 1 and Figure S1, would be estimates of a single, continuous spline for time relative to sibling birth. It's hard to know from eyeballing it, but my guess is a jump in cortisol would still be apparent, for example, but it would be more gradual and less striking than the current figures suggest.

We thank the reviewer for highlighting this insufficient communication in the previous version of the manuscript. Indeed, what we primarily did was a model comparison between a model with a continuous smooth only and a model with the same continuous plus the categorical before/after variable allowing for additional intercept differences between the time before and after sibling birth. We have now made this point much more explicit in the methods and the Results section.

It might be important to emphasize here that we did not calculate separate smooths for before and after sibling birth, to avoid the associated pitfalls and also misleading statistical results, also since predictions would still be calculated ( = extremely extrapolated) for the entire time period. Therefore, we decided to use the cumulative nature and allow the model to estimate both a continuous smooth for the entire time period plus an intercept difference in values. This approach comes at the small inconvenience that the model allows only for discontinuity of the predicted values but not the first and second derivatives, i.e. there will be no discontinuity in the slope of the smooth at sibling birth. Although this leads to a slight constraint on optimal smooth estimation (most pronounced in Figure 1A), we think that this is the best possible approach.

Line 213: Is this Chi-squared value supposed to be negative?

No, thank you for highlighting this mistake. We checked and corrected now throughout the manuscript.

Discussion: My public review mentions that it is difficult to interpret the finding of increased cortisol following sibling birth. At some points, the authors appear to recognize this, as they raise a number of good arguments in the discussion about potential factors leading to a TTS-timed cortisol increase. Some, like a sibling birth coinciding with the beginning of juveniles experiencing greater male aggression, have nothing to do with TTS per se. Other arguments imply a role for beneficial behavioral interactions between the older and younger sibling.

We thank the reviewer for highlighting this. We have now added other factors that may influence cortisol changes in the older offspring. (lines 389-408)

Yet, the authors appear to dismiss those concerns and conclude that the cortisol increase indeed implies stress. First, I recommend at minimum addressing why numerous alternative interpretations-some that the authors identify, and some that I have identified-should be disregarded in favor of the "cortisol = stress" conclusion.

Throughout the manuscript we claim that cortisol is a stress response or indicating a homeostatic challenge. As alternative interpretations we tried to exclude that the found increase in cortisol is related to food stress challenges (urinary total T3). We further found that the proxy for the function of the immune system (urinary neopterin) was affected. With this two finding we came to conclusion that the homeostasis of the youngsters must be challenged by a non-food related stressor, but in a way that is affects the immune function.

We now added in the introduction that cortisol is released to a threat of homeostasis. We avoided to use the world “stress” without any further description. We also changed the title.

Second, as I state in the public review, having data on behavioral interactions thought to be stressful would go a long way towards solidifying the interpretative elements of this paper. I don't know whether this sort of information is available, but if it is, I recommend integrating it into a revision of this manuscript. One of the main lessons from the developmental psychology literature is that there is substantial heterogeneity in TTS adjustments, which may be attributable to family dynamics or larger ecological factors. Being able to test these kinds of theories in bonobos would significantly boost the impact of this paper.

Unfortunately, we do not have data describing behaviour in detail during TTS in wild bonobos. We have now added:

“To clarify that our study is describing a physiological stress response but only discuss the potential reasons for the response we add: Whether the increase in cortisol levels in the bonobos in our study had positive or negative long-term consequences was not assessed. Future studies should therefore integrate behavior and physiological measures to estimate the impact of TTS for the older sibling. Such a combination of measures might help to disentangle why young bonobos show such an intense cortisol response, for example as a response to the novel mother-infant constellation, as a measure of positive valence arousal, or with TTS as a normative maturing experience. (lines 452-458)”.

Line 304-305: The authors claim that "findings from human children show behavioral responses to sibling birth [are] independent of their actual age". True, some show that, but others do not; many claim younger children exhibit larger behavioral disruptions (e.g. Nadelman and Begun, 1982; Volling, 2012). Some expansion on why age might matter in some aspects of adjustment, but not others, would provide useful context to the discussion.

We have now in cooperated these thoughts.

“However, changes at sibling birth can be age dependent. In response to sibling birth, scores for e.g., clinging and other gestures of reassurance were negatively correlated with the age of the older sibling (Dunn et al., 1981; Nadelman and Begun, 1982; Volling, 2012). Thus, in children, age seems to affect the behavioral response towards, or the perception of, the arrival of a sibling.” (lines 381-385)

Figure 1: The right panels of Figure 1 (B, D, and F) will be very hard to interpret for the average reader: the figure captions are sparse, the contours have labels that are so small as to be invisible, no justification/explanation is given for the extrapolation parameter (and thus it won't be clear to the vast majority of people why the figure is splotchy), etc. I understand the objective of contour plots for visualizing non-linear interactions, but as they stand the figures and/or captions need to be changed to form a self-contained explanation, as figures should. I leave edits up to the authors' discretion: they could do some combination of beef up their figure captions; present a more traditional interaction plot of linear effects (i.e., present marginal trends of -1, 0, and +1 SE on offspring age), since all the splines in Figure 1 are very close to linear; or present a different kind of plot, like a perspective plot, or a plot of predicted splines using the get_predictions function in itsadug.Figure 2: I think the caption for supplementary Figure 2 is incorrect-there is no depiction of time relative to sibling birth in this figure, just the age of the older sibling. It appears the caption was incorrectly copied over from part of Figure 2.

We have now – also following rev1 – revised all figure captions to improve the understandings and self-standing of the figures.

Data: The DOI linked for the source data doesn't work. I would need to re-evaluate the paper after having access to the raw data, so I can verify the analytical reproducibility of the key claims.

We are really sorry and apologize that the doi link was not functioning. We have now solved this issue.

Reviewer #3 (Recommendations for the authors):Abstract:I would not identify the birth of a sibling as a developmental transition. Perhaps would be better to say a major life transition or major transition in the early life of an individual. Or if you stick with the word development, perhaps it would help to elaborate a bit to make clear why it is a developmental transition. Maybe it applied to nutritional development so that the mother-infant nutritional relationship must end with the birth of a new sibling? Perhaps you are referring to social development? Elaborating a bit might make clearer what you mean.Should be: "Studying the transition", "in the mother-infant relationship".

We thank the reviewer for the suggestion. We have now changed the sentence to:

“In animals with slow ontogeny and long-term maternal effort, immatures are likely to experience sibling birth before reaching maturation. The birth of a sibling marks a major transition in early life, maternal investment is constrained, and it deprives the older offspring from maternal support.” (lines 30-33)

Change to "evolutionarily old"- although evolutionarily old is a bit vague. Perhaps you can narrow it down to a specific time based on comparisons with other primates or mammals. For instance, you might say that this effect was likely present in the common ancestor of all the great apes. Something more specific to your study.

We have now speculated: Our results suggest that bonobos and humans share the syndrome of TTS and that this developmental stage may have emerged in the last common ancestor. (lines 42-44)

Line 43: change "while still being dependent" to "while they are still dependent".

We thank the reviewer for the suggestion. This part does not exist anymore.

This sentence also needs a reference- also do you mean children in humans only? I guess that is what you mean but perhaps you should extend this to all primates or long-lived, slow developing mammals. You could say that in mammals with slow rates of growth and development that give birth to single offspring at a time, most offspring are exposed to the birth of a younger sibling.Extending this to other species could make it more relevant to more researchers.

We thank the reviewer for that great idea and started the introduction with:

“In animals with slow ontogeny and long-term maternal effort, immatures are likely to experience sibling birth before reaching maturation. The birth of a sibling marks a major transition in early life, maternal investment is constrained, and it deprives the older offspring from maternal support. Transition to siblinghood (TTS) is often considered to be stressful for the older offspring, but physiological evidence for this is lacking.” (lines 30-34)

Line 45: competitor in what sense? Competitor for maternal attention and resources, or for food in the environment, or for reproduction or for social partners etc. I think this needs to be more specific.

We have now added that it is about resources in general. (line 72)

Line 49: Do you mean behaviors shown by the older sibling? This is not clear. Make clearer that it is the older sibling who could show increased aggression, clinginess, and depression. Otherwise, could be perceived to be in the mother or in the new sibling.

We now wrote: “Accordingly, in humans, TTS is considered to be a stressful life event or even a disruptive crisis for the older sibling even under favorable conditions, a perspective that seems to be supported by TTS-related behaviors of the older offspring such as aggression, clinginess, and depressive syndromes (reviewed in Volling, 2012; Volling et al., 2017).” (lines 74-77)

Line 51: not sure what you mean by mere behavioral adjustments. I think the point you are making here is important, but couldn't behavioral adjustments be in response to stress. I guess you are trying to say that these are behavioral changes that could occur in response to new siblings that are not associated to stress for the older sibling. Make clearer.

We have now – also following the suggestion of reviewer 1 changed it to: “However, whether behavioral changes during TTS are associated with sibling birth rather than age-related behavioral adjustments related to withdrawal of maternal support remains to be resolved (Volling, 2012; Volling et al., 2017).” (lines 79-81).

Line 94-97: I do think this statement needs references to support it.

We have rewritten this part.

Line 98: I think this statement also needs some references and a range in inter-birth interval lengths for bonobos, including a variety of sites, since you have the ones for your site specifically, below.

We have now included the range of inter-birth intervals of wild bonobos, also including other sites.

We leveraged the large variation in inter-birth intervals in bonobos ranging from three to nine years (Knott, 2001; Tokuyama et al., 2021) to differentiate between the effects of TTS on one hand, and nutritional and social weaning on the other. (lines 136-138)

Line 101: should be social dependency on the mother.

This was also part of the re-written part.

Line 106: would co-dependency on the mother be better than co-residence? Co-residence sounds like one offspring needs to leave the natal group at some point, which is not always the case. Co-residence with siblings can occur for their entire lives if they are the philopatric sex.

We have now changed it to: “Maternal support is intense and persists for a long time (Stanton et al., 2020; van Noordwijk et al., 2018), and extended periods of parental care of two dependent offspring of different ages is common (Achenbach and Snowdon, 1998).” (line 100-102)

Line 110-111: Sexual maturation at 4 years of age is very young… Many individuals might still be nursing at 4 years old. Perhaps your definition of the onset of sexual maturation needs to be explained. Also, unclear how the onset of sexual maturation is different from the onset of menarche, and how these two milestones are different, so this should be cleared up as well.

This part is now re-written.

Introduction in general:You've made many comparisons between bonobos and humans, which is good, but I wonder if it would be good to add in data and comparisons with other apes as well (chimps, gorillas, orangutans). I know you mention chimpanzees once or twice, but I wonder if it would help to make more explicit comparisons throughout with the other great apes. I say this because eventually in your discussion, you are going to be making some preliminary conclusions about when various developmental and stress mechanisms would have evolved and saying that the timing would have been in the last common ancestor between bonobos and humans may be incomplete. Inherently we would want to hear about the chimpanzee data at least, to know if we should think about the last common ancestor with Pan, but of course we would wonder about any evidence in the other great apes. If the other great apes show similar patterns in life history, development and maternal investment that you are pointing out for humans and bonobos, then the findings of your research here could also apply to them (pending studies in these other apes). Alternatively, there might be some key differences in the life history patterns and infant development of different apes, so it would be good to know about them to later understand why you may not think that the findings extend to a certain genus.

We have now added information about sibling interaction or the effect of sibling birth in other apes (lines 102-119)

I suggest you try to keep the tense for your article in the past tense as much as possible. As it is now, you switch quite a bit between present and past tense, and at times, you use present tense but this is a bit awkward. Past tense would be best.

We now carefully checked to have the manuscript in past tense. We apologize if there are still mistakes.

Line 123: You should mention the name and place of your study population here since you are bringing it up to set up your study.

We have now added: “Our study investigated TTS-related changes in physiological responses in juvenile individuals of two habituated groups of wild bonobos (Pan paniscus) at the site of LuiKotale in the Democratic Republic of Congo.” (lines 131-133)

Line 125: this statement on the dependency levels of the offspring needs to be supported with a reference and it needs to be quantified in some way. Is dependency based on nipple contacts, proximity, sleeping in the same nest, being carried from place to place, all the above?Also, unclear if this statement refers to past research or if this is something that is being done in this paper. Did someone already investigate the nutritional development of these individuals, or is the level of dependency something you will be measuring in the present study and measured relative to TTS? This needs to be made clearer.

Also following reviewer 1 we have now included definitions and this was the case in our study.

“… and the state of the older siblings at the time when mothers gave birth to another infant ranged from highly dependent (i.e. infant would die without mother) to rather independent (i.e. offspring would likely survive without mother).” (lines 140-142)

Lines 139-140: I think this sentence needs some references. Also, you explain T3 in greater detail but not the neopterin hormone. I think a sentence specifically on neopterin and how it works or what it means can help here.

Following also suggestions of reviewer1 we have restructured the whole paragraph and have added more information on neopterin and references (lines 147-169).

Lines 140-141: I am not quite understanding T3 can help disentangle effect of energetic stress versus social stress. Can you please make this more obvious/clearer?

Following also suggestions of reviewer1 we have restructured the whole paragraph and have added more information about T3 and energetic deficiency (lines 164-169)

Line 143: unclear why neopterin levels should decrease because we do not have enough information about neopterin.

We apologise for being less clear. We hope it is now better explained better with the changes we added (lines 156-163)

Line 146: you say nursing here but later you define suckling as contact with the nipple. Might be better to stick with one of the two terms, especially since people are going to keep an eye out for the term nursing to see how this was defined later in the text.

We have now changed nursing to suckling (line 170).

Line 148-150: Are you saying that social weaning is weaning from the mother more generally, in terms of proximity and access to carrying, etc.? This is very confusing because weaning in most infant development/maternal investment literature refers specifically to nursing or feeding behavior. Until this point in the article, I was thinking you were trying to make a distinction between nutritive and non-nutritive nursing.If you would like to talk about changes in the mother-infant relationship outside of nursing and foraging behavior, I strongly encourage you NOT to use the term weaning for that. Perhaps switch to "attainment of physical independence" or some other similar phrase since the only other measures outside of nursing behavior that you are looking at are more physical measures (so the infant being carried versus moving on its own/ infant in certain proximity to the mother). Instead of nutritional and social weaning, you could say weaning and attainment of physical independence.

We are grateful for this suitable wording. We have now changed the terms to weaning and attainment of physical independence. Many, many thanks for the wording.

Line 193: With increasing age, urinary…Also, you say that they significantly changed in males but not females, but then you say that this change was not significant… I am confused.

We have now re-worded the result section to be more clear and precise following also suggestions by reviewer2.

Line 210-211: same issue here. you say significantly declined but then in brackets you said sex difference not significant. Which is it?

See above, now it is re-written, and not contradictive anymore.

Line 75, 262 and elsewhere: You keep talking about nutritional versus social weaning (or as others have said, nutritional versus behavioral weaning), but this concept is fairly recent. The idea that there are two components to weaning, the milk-transfer and the psycho-social relationship between the mother and the infant through continued nipple contacts, regardless of whether or not milk transfer occurs, is a fairly new and innovative idea that needs to be supported with literature in your paper, and that needs to be explained a little bit somewhere in the text. For example, research on wild chimps and other primates has shown that comfort nursing, without milk transfer, can occur for years after lactation has ended. This sets up a situation where you have these two separate weaning periods: weaning from milk and weaning from nipple contact- so the nutritional versus social/behavioral weaning. Thus, the mother-infant behavioral relationship can develop separately from the mother-infant nutritional relationship, despite considerable overlap between the two. This distinction, and its importance for the infant and mother, should be explained.

We thank the reviewer for these thoughts. We have now added:

“The term weaning is often used for the attainment of nutritional independence, but also comprises the process of social independence and behavioral maturation, which can occur at different ages. Nutritional weaning refers to the termination of the consumption of maternal milk, but nipple contact of the offspring without milk transfer may exceed lactation and is assumed to be social comfort behavior (Bădescu et al., 2016; Berghänel et al., 2016; Matsumoto, 2017).” (lines 85-90)

Line 276-279: Are you saying that TTS is severe, uncontrollable for older offspring, and was unpredictable or moderately unpredictable? Make clearer how this statement relates to your results.

Yes, this was indeed what we think that could be the scenario. We reworded the sentence to:

“The intensity of a stress response is generally determined by the severity, controllability, and predictability of the stressor (Seiler et al., 2020), all of which probably apply to TTS as a stressor, being novel, sever, uncontrollable and relatively unpredictable for the older offspring, and therefore contribute to the comparably high cortisol response that we observed in our study.” (lines 337-341)

Line 347-349: change sentence to"However, strong negative effects caused by a highly predictable and normative stressor that invariably affects most individuals, such as the birth of a sibling in apes, should be under negative selection, and would thus be expected to XX XX (explain what this would mean)."

We thank the reviewer for the wording, now we added:

“Moreover, strong negative effects caused by a highly predictable and normative stressor that invariably affects most individuals, such as the birth of a sibling, should be under negative selection, and would therefore be considered to be a non-adaptive trait.” (lines 442-444)

Line 298: I would change this to"…suggest that nutritional weaning is usually completed by around 4.5 years of age".

We have now changed the sentence to: “…, and preliminary analyses of stable isotopes in samples collected from the same population suggest that nutritional weaning is completed by around 4.5 years of age (Oelze et al., 2020).” (lines 372-374)

Lines 372 to 372, or more broadly for this whole paragraph: Similar to my comment for the abstract, the statement that your results highlight the evolutionary history of stress response and TTS is vague and almost seems like a throw away idea because it is not specific enough. Can you go a step further and talk about the possible timing of an evolutionary link between stress and TTS? Presumably, this important interaction would have appeared with the great ape transition? Or would it be with the transition between genus Pan and Homo? Use the comparisons in life history traits and infant development of humans and the other great apes to infer when this mechanism that you found could have become more important or more prominent.

We have now added a speculation about when TTS might have occurred (last paragraph of the discussion) (lines 474-491)

Methods:Line 397: I would like to see a table with a breakdown, by infant age and sex, of the sample sizes and numbers for the behavioral and urine data.

We agree that a table is helpful, bit while the data are not presented by individuals age, we thought it might be helpful to present them in relation to sibling birth. We now added a table in the supplement presenting the data in relation to sibling birth. If the age of the bonobo is still important, we can change the table or the reader might use the raw data available. (line 504, supplement table 1)

Lines 399 to 411: Did you exclude from the calculations time out of view, or time when the infant was ventral, but it was unclear if they were in contact with the nipple, and excluded time in a nest with the mother since you can't see the infant? If yes, say so, if not, justify why you did not.

Yes, we excluded all data points where the infant was not sufficiently visible, and also state this now in the methods section.

I am not familiar with any of the urine analyses so cannot comment on these.Please explain how the proportions of the behavioral measures were calculated somewhere in the methods.

Has been added to the methods section (line 515ff, especially line 526f).

[Editors' note: further revisions were suggested prior to acceptance, as described below.]

The manuscript has been greatly improved, but there are some remaining issues that need to be addressed before it can be accepted for publication.– First, and most importantly, I (the reviewing editor) strongly urge you to consider the comments of reviewer #2 w/r/t the plausibility of the two-intercept model. While I understand that this model provides the best fit statistic, in my opinion, fit statistics should not override biological plausibility. There is no biologically plausible explanation for why cort would suddenly drop at 7 months (or at least, none that have been relayed in this text), and basic data visualization -- i.e., the raw scatterplot of the data--does not suggest this is what is happening. I personally feel that the single-cut discontinuous spline is a good middle-ground choice (due to the suddenness of sibling birth), followed by a continuous spline, followed by the two-cut discontinuous. The last two options would certainly be worth including in the supplementary materials.

We have now focused our results on the cortisol and neopterin model with one cut at sibling birth in the results as well as in the discussion. We have now the three physiological markers with one cut as the main figure 1. As suggested, we now added the models and the figures for cortisol and neopterin with the second, unexpected sudden change, as supplement material. We further have now reworded also the part in the result section to already highlight that the results are unexpected and that we do not have an obvious explanation for.

This issue is the primary reason that I am recommending a revise and resubmit, rather than an acceptance. Please note that my acceptance of the manuscript is not contingent upon making this change. I respect that authors may have different points of view surrounding the interpretation of fit statistics. However, if it is not changed, the final evaluation summary will reflect the fact that myself and one of the other reviewers disagreed with the analysis strategy. I want to be transparent about the source of the disagreement so that the evaluation summary would not come as a surprise should you choose not to change which model you prioritize in the main text.

We thank the editor for her honest words. We could understand the argumentation of reviewer2 and the editor, and therefore, made the adjustments as described above.

– Please include a scatterplot of the raw data in the supplementary materials.

We have now included a scatter plot of the cortisol data in the supplementary materials, Supplementary Figure 3.

– Reviewer #2 is correct that though the language around 'stress' has been considerably improved, there are still places where there are ambiguities in how it is used/what it is implying. The manuscript will be stronger with these ambiguities removed. These data are quite interesting and do not need to lean on 'stress' in order to be noteworthy and important. Similarly, I agree with reviewer #3s concerns about characterizing TTS as a developmental stage. Both of these are wording issues that should be simple to resolve.

We have now carefully re-worded the MS. Describing sibling birth as an event and cortisol changes as cortisol changes.

– Please follow reviewer #3s request for a thorough proofread. There are a considerable number of spelling and grammar mistakes that remain, which is distracting as a reader. They have many helpful suggestions for places where the language could use additional clarification, which again, will help strengthen the final product.

Alison Ashbury (see Acknowledgement) edited the manuscript.

Reviewer #2 (Recommendations for the authors):Most of the issues I raised have been sufficiently addressed. Most prominently, the data and code are now available, and I was able to check that the authors' results are reproducible (Side note: I recommend further commenting the code so it is clear e.g. which models correspond to which figures, and I also recommend sharing the code in a different format than a Word document-these steps would help with accessibility).

We thank the reviewer for the suggestions. We have now uploaded the code as txt files. We will write additional comments on the final code, if we all agree for example on figure order and files.

I have a few remaining concerns:Lines 75 – 78: "Accordingly, in humans, TTS is considered to be a stressful life event or even a disruptive crisis for the older sibling even under favorable conditions, a perspective that seems to be supported by TTS-related behaviors of the older offspring such as aggression, clinginess, and depressive syndromes (reviewed in Volling, 2012; Volling et al., 2017)."As I wrote in my initial review, this claim is an oversimplification of these papers. Volling (2012), in particular, clearly concludes that there is substantial evidence for either of two quite different perspectives: that TTS is stressful, OR that it is an occasion for ecological adjustment that does not manifest in stress. This section of the intro should be edited to better reflect the ambiguity of the background literature, and the many different ways in which "stress" might be operationalized (see also my last comment below).

We absolutely agree, we have now added in the introduction:

“However, sibling birth also presents opportunities for the older offspring, such as social and emotional growth through interacting with the newborn. Individuals vary in how they adjust to the birth of younger sibling; some children have difficulties while others cope well (reviewed in Volling, 2012; Volling et al., 2017). In any case, the birth of a sibling is linked to a time of change the older child must cope with.” (lines 75-79)

Lines 360 – 361: "The sudden recovery of cortisol and neopterin levels to pre-sibling birth levels after seven and five months, respectively, is puzzling and requires explanation."I appreciate the authors expressing some caution here, but I would go further, which raises a somewhat larger point. The authors' statement I quoted is only warranted if the "two-cut" discontinuous model is taken at face value, and I am skeptical that should be done. The scatter plot of cort values, ignoring any splines, shows an apparent increase in the period right after sibling birth. That's interesting and worth trying to understand. But there are plenty of cort values just as high right before and right after the post hoc seven-month window. The "sudden-ness" of the increase and decrease is a function of specifying different intercepts at two different points. I wrote in my last review that I favored a single continuous spline for figures. The authors have basically responded by saying that "a one-cut discontinuous spline is a better fit than continuous, and a two-cut model is better still." But we have to consider biological plausibility, not just fit statistics. Do we really think it's realistic to expect such a uniform and sudden change seven months in? If we consider plausibility, I think one can make a reasonable case for a continuous spline, or one with a discontinuity at sibling birth (which is a prominent and biologically meaningful event), but not so much for the "two-cut" model. I recommend the authors foreground more biologically plausible models, and if they really still wish to include the "two-cut" model, that it occupies a less prominent position.

We agree with the view that the second cut is unexpected and without an obvious biological expectation. We have now focused our results on the cortisol and neopterin model with one cut at sibling birth in the result section, because we agree that we here had an event.

Figure 1 which was before showing the two-cut model is now the three physiological markers with one cut. As suggested by the editor, we added the models and the figures for cortisol and neopterin with the second unexpected sudden changes as supplement material. We further re-worded also the part in the result section to already clarify that the results are unexpected. We further added the cortisol scatter plot for the reader as supplementary material as suggested by the editor.

“While the model with the two discontinuities describes our data better mathematically, there is no obvious biological explanation for the second change (i.e., the sudden decline in cortisol) after seven months. However, in the model with only one discontinuity at sibling birth and a smooth continuous decline thereafter (Figure 1A-C), the cortisol levels took over 7 months to return to previous levels.

Hence, the absence of low cortisol levels after sibling birth was evident in both models.” (lines 217222)

Lines 488 – 490: "Yet, the results obtained from wild bonobos renders support to the long-standing but untested and recently questioned assumption that the birth of a sibling is a stressful event for the older offspring (Volling, 2012; Volling et al., 2017)".Once again, how are we to conclude that a cortisol increase equates to an individual experiencing a "stressful event"? The authors have become more careful in their language in many parts of the manuscript, but not here. The authors might say that they are simply using "stressful" to mean "a deviation from homeostasis", but my point all along has been that the authors need to be very, very clear about operationalizations if they want to use the term "stressful" in a different manner than its widely understood everyday definition. I recommend being super explicit about the usage and definition of the term "stress" in the intro, as I mention above. I mean literally provide a clear definition, upfront, so all readers are on the same page. The brief link between cortisol and homeostatic functioning, which isn't given until p. 5, is too indirect and vague. I also recommend removing statements like the one I quoted above. The authors already state in the discussion what we can actually determine here: cortisol goes up after the birth of a sibling, and it doesn't seem to be related to their age when it happens. Whether this event is "stressful" or not (which the authors claim without contextualization) cannot be determined by this dataset.

We have now added in the introduction that we expect an increase in cortisol as a response to the challenging situation of sibling birth. We also re-formulated many sentences in the discussion. E.g., lines 78-79, lines 151-153. Line 362, lines 347-341, line 352 and so on.

Reviewer #3 (Recommendations for the authors):I am satisfied with the changes made by the authors based on my first review.General comment: I would suggest a thorough proof-read of the text, just to make sure there aren't any little vocab or grammar mistakes and to polish it up. I caught many of the mistakes and pointed them out in my review, but I'm sure I missed some.

We thank the reviewer for all her helpful comments and corrections so far. We had sent out our manuscript for proof reading, and we think it sounds now way better. We hope that the reviewer agrees.

Title: would be better to say "causes a substantial and long-lasting increase in…"?

We have now changed the title to: Transition to siblinghood causes a substantial and long-lasting increase in urinary cortisol levels in wild bonobos

Abstract:Second sentence could be worded better, as it is hard to understand. Perhaps "In these species, the birth of a sibling marks a major transition in early life, as maternal investment is constrained, and older siblings experience a decrease in maternal support."

We have now changed it to: In these species, the birth of a sibling marks a major event in an offspring’s early life, as the older siblings experience a decrease in maternal support. Lines 31-33

Following sentence could be worded better:"In the older offspring independent of its age, with siblings' birth, urinary cortisol levels increased fivefold and remained elevated for seven months."Maybe better would be: "Following the births of siblings, urinary cortisol levels of older offspring, independent of their age, increased fivefold and remained elevated for seven months."Change "did not show corresponding change" to "did not change". Otherwise, it is confusing for a few reasons: first, should be changes, second, make one wonder what corresponding means. I suggest you just simplify and cut out the unnecessary "show corresponding" part.

We have now changed it to: The cortisol level increase was associated with declining neopterin levels, however T3 levels and behavioral measures did not change (lines 39-40).

I am not a fan of the word syndrome. Sounds like a disease or physical problem. Could you say:"Our results suggest that bonobos and humans experience TTS in similar ways and that this…".Also, in this sentence, I do not like the idea of TTS as a developmental stage. I mentioned this in my last review and the authors made the change, but perhaps the idea got lost in the revising and then it was added back in here. TTS is more of a life stage or a life history stage… or a life transition. Would you say that offspring whose mothers never make a sibling are underdeveloped because they were not able to undergo the developmental stage of TTS? No because TTS is not a developmental stage, like weaning is, for example. So, I would reframe the idea of TTS that does not imply that it is a stage of development.

We absolutely agree, we have now changed it to: Our results suggest that bonobos and humans experience TTS in similar ways and that this developmental event may have emerged in the last common ancestor (lines 442-43).

Line 50-51: but maternal provisioning through food sharing and feeding of young non-milk foods occurs after weaning. Revise this sentence to make clearer you are talking about milk and/or nursing.

We have now revised the sentence to: In mammals, weaning refers to the transition from nutritional dependency to a stage when immatures are independent of maternal food provisioning (lines 49-50)

Line 64-66: This sentence should be simplified and cleaned up. It is unnecessarily complicated and wordy, right now. Also, what are you trying to say exactly? That there are negative effects associated with having to share maternal care? Be more specific.

We now changed it to: However, the older offspring must share maternal care, which may influence its social behavior as well as its physiological constitution (lines 62-63).

Line 69: Maybe change mother-offspring dependency to something less extreme, since dependency almost sounds like the offspring will be completely reliant on the mother for its whole life.

We have now changed it to something less extreme: Immatures grow slowly, social maturation extends well into adulthood, and to a certain degree, beneficial mother-offspring relationships can last a lifetime (lines 65-67).

Line 91: change to "to be a social comfort behavior"Also, the Badescu reference should be 2017, as it came out in early view in 2016 but then received an official issue in 2017. It is always very confusing.

We are sorry for the wrong citation. We now changed it to Bădescu et al. 2017 in lines 89-90 as well as in the references (lines 720-722).

Line 107-108: the last part of this sentence may not be grammatically correct.

We have now changed it to: While data on mother-offspring relationships are abundant, little is known about interactions between immatures and infants born to the same female (lines 101.103).

Line 137: Not sure what constitution means. Can you replace with a more specific term?

This part was changed during editing to: We used multiple physiological and behavioral measures to investigate the responses of older siblings to the birth of their younger sibling. We sought to disentangle the effects of changes in mother-offspring relationships and energetics that are associated with nutritional and social weaning, from the specific effects of a younger sibling’s birth (lines 129-133).

Line 144-145: Hmm. I guess I understand what you mean but is this something measurable, that you determine from certain variables? It seems a bit speculative to say assume that the infant would die or would survive. Perhaps you can qualify these statements by offering as examples the variables you used to determine this. For example, you likely mean nursing behavior. So, the highly dependent were still regularly nursing, for prolonged periods, whereas the independent hadn't been seen nursing X focal hours of observation. Or maybe you mean the time that infants spent in body contact. Whatever you used to identify independence, give us that information.

We have now added this information: … and thus the developmental status of older siblings at the time when their mothers gave birth to another infant ranged from highly dependent in terms of travel support and foraging skills (i.e. time carried and nursed) to mostly independent (lines 136-139).

Line: Visual inspection of?

We added now: Visual inspection of the data showed that the sudden decline in riding at sibling birth was only evident in older siblings belonging to the younger age cohort… (lines 285-287).

Line 194: like here for example, should be "were" not "was". A small error but there are several like this throughout the text so go through carefully to polish up the writing.

We thank the reviewer for detecting all the little mistakes. We hope that the others we all changed by the editing service.

Line 334: Do you mean "or showed a sudden change but in the opposite direction…"?

The sentence was completely revised to: At sibling birth, weaning-related behavioral changes were either already completed (independent foraging and suckling), did not change discontinuously (urinary total T3, suckling, time in spatial proximity to mother, and independent foraging), changed suddenly in directions opposite of our expectation (increasing body contact time with the mother), or were significant only in subjects belonging to the younger age cohort (riding) (lines 332-337).

Line 340-341: levels returned to the levels before within one day. This phrase could be worded better or constructed a bit better to make easier to understand

We have now changed it to: A similar cortisol response occurred in bonobos in response to a group member giving birth, but in this case, the individual’s cortisol levels returned to previous values within one day (lines 340-342).

Line 361: Wait, why is this puzzling. You just told us in the previous sentence that it takes human children 8 months to recover after sibling birth. Then the recovery you found after seven and five months is right in line with what you expect. Why puzzling?

We have now explained – hopefully better – that it is not puzzling that the levels take a long time to decline, what is puzzling is that the model with the two cuts would suggest that cortisol levels decline at a time suddenly, similar to the sudden increase. We do not have an obvious explanation for why they should suddenly decline. When levels suddenly increase the sibling birth occurred, however, nothing we are aware of happens after 7-month period.

Line 378: Unless I missed this somewhere, it would be good to add a note about whether some of the siblings continued to make nipple contacts after the birth of a new baby. And maybe tell us how many of the total number of infants were observed to continue nursing after birth of baby. This is interesting information for others. Maybe in results would be good to add?

In the results we had written: The proportion of time the older offspring was observed in nipple contact showed a continuous decrease prior to sibling birth in both males and females, and reached zero about two months before sibling birth (Figure 2A-C, Table 3) (line 266 and 268). We hope these answers the question.

Line 382: Say what kind of samples. Fecal samples? Hair samples? Urine samples? Bone samples?

We now have added fecal (line 383).

Line 403: Affiliative intends? Not sure what you mean here.

We have now changed it to: intentions, as suggested by reviewer 2 (line 403).

Line 449: In baboons'? Maybe in baboons,…? Something is off here. Also, baboons repeats a few times. Some errors need fixing

We have now changed it to: For example, the presence of a sibling did not affect the HPA-axis later in life in baboons, but other early life adversities had lasting consequences (lines 448-449)

Line 486: I don't like the word syndrome in this context. Makes it sound like TTS is a disease.

We have now changed it to: physiological changes (line 484).

In methods, might be good to mention which authors collected which data, which authors did which analyses, etc. For example, the behavioral data were collected by focal animal sampling. Was this done by all the authors together, or just one? Give the initials of the author(s).

We thank the reviewer for this suggestion; however, this information is provided in the authors contribution.